# Bacterial vitamin B6 is required for post-embryonic development in *C. elegans*
Min Feng[1], Baizhen Gao[1], Daniela Ruiz[1], Luis Rene Garcia[2] & Qing Sun [1]✉

Nutritional intake influences animal growth, reproductive capacity, and survival of animals. Under nutrition deficiency, animal developmental arrest occurs as an adaptive strategy to survive. However, the nutritional basis and the underlying nutrient sensing mechanism essential for animal regrowth after developmental arrest remain to be explored. In *Caenorhabditis elegans*, larvae undergo early developmental arrest are stress resistant, and they require certain nutrients to recover postembryonic development. Here, we investigated the developmental arrest in *C. elegans* feeding on *Lactiplantibacillus plantarum*, and the rescue of the diapause state with trace supplementation of *Escherichia coli*. We performed a genome-wide screen using 3983 individual gene deletion *E. coli* mutants and identified *E. coli* genes that are indispensable for *C. elegans* larval growth on originally not nutritionally sufficient bacteria *L. plantarum*. Among these crucial genes, we confirmed *E. coli pdxH*, and the downstream metabolite pyridoxal 5-P (PLP, Vitamin B6) as important nutritional factors for *C. elegans* postembryonic development. Transcriptome results suggest that bacterial *pdxH* affects host development by coordinating host metabolic processes and PLP binding. Additionally, the developmental arrest induced by the *L. plantarum* diet in worm does not depend on the activation of FoxO/DAF-16. Altogether, these results highlight the role of microbial metabolite PLP as a crucial cofactor to restore postembryonic development in *C. elegans*.

A well-balanced diet contains macronutrients such as carbohydrates, fats, and proteins, as well as micronutrients such as vitamins and minerals to sustain basal metabolism and tissue function. Lack of certain nutrients leads to alterations in metabolism and causes severe deficiency in animal growth and reproduction. Therefore, understanding the role of individual nutrients is critical for revealing regulatory mechanisms of animal development.

The nematode *Caenorhabditis elegans* is a good model to study the effects of diet on life history traits such as development, reproduction, and aging[1]. As a bacterivore, *C. elegans* consumes a wide range of bacteria, including bacteria found in human microbiota such as *E. coli*. Multiple genetic engineering techniques have been developed to modify the genome of both *C. elegans* and its diet bacteria, making this model suitable for investigating the effects of bacteria-derived nutrients on host development[2]. Nutrient availability is a determinant of larval development in *C. elegans*. To survive unfavorable nutritional environments, the larvae can alter their developmental progression[3]. For example, in the absence of nutrients, newly hatched L1 larvae immediately arrest their postembryonic development[4]. Therefore, the *C. elegans* developmental arrest model provides an opportunity to study how the animal adapts to fluctuations in nutrient availability.

Animals obtain essential nutrients not only from food intake but also from the synthesis of co-existing symbiosis microbiota. Previous studies emphasized the importance of metabolites derived from bacteria on larvae development in *C. elegans*. For example, vitamin B12, a metabolite of *Comamonas aquatica* D1877, accelerates host development via the methionine/S-Adenosylmethionine (SAM) cycle[5]. In addition, bacteria-secreted enterobactin facilitates mitochondrial iron uptake and promotes worm development by binding to the ATP synthase[6]. Additionally, vitamin B2 provided by bacteria regulates worm protease activities and development[7]. However, the effects of microbial metabolites/nutrients on postembryonic development in animals are missing, which is essential for understanding both the nutritional requirements and the intricate mechanism involved in animal growth.

In this study, an interspecies systems biology approach with *C. elegans* and its microbial diets, *E. coli* and *Lactiplantibacillus plantarum*, were used to identify bacterial metabolites that affect the animal's early-stage development and gene expression. *L. plantarum* has been shown to have probiotic effects in *C. elegans* when fed to adults[8-10]. We observed that hatched L1 *C. elegans* developmentally arrest or stalled when feeding

[1]Department of Chemical Engineering, Texas A&M University, College Station, TX, USA. [2]Department of Biology, Texas A&M University, College Station, TX, USA.
✉e-mail: sunqing@tamu.edu

on *L. plantarum* diet; however transient feeding with trace *E. coli* can promote larval development and further feeding on *L. plantarum*, which was originally not nutritionally sufficient. Through a systematic screen on ~4000 single gene deletion strains, we identified 29 *E. coli* genes that are essential for *E. coli* to promote *C. elegans* growth on an *L. plantarum* diet. The *E. coli* pdxH gene and the downstream metabolites vitamin B6 were further confirmed to coordinate host metabolic processes and PLP-binding activity thereby contributing to the host development.

## Results

### *L. plantarum* induces developmental arrest in *C. elegans*

The nutritional composition of bacterial food can induce larval developmental delay in *C. elegans*. To investigate the effect of two different food sources, a commonly consumed probiotic Gram-positive bacterium *L. plantarum* and the standard laboratory food *E. coli* OP50, we fed synchronized L1 worms with *L. plantarum* and *E. coli* OP50 individually and measured worm development. Synchronized L1 worms were fed with *L. plantarum* or *E. coli* OP50. We found that ~95% of worms fed on OP50 developed to adults within 3 days at 21–22 °C, whereas worms fed solely on *L. plantarum* developmentally arrested at the early larval stage (Fig. 1a), suggesting that *L. plantarum* may not provide sufficient nutrients for *C. elegans* larvae development. To further investigate the food preference of *C. elegans*, two behavior assays to assess food dwelling and food choice assay were performed. Briefly, in the food-dwelling assay, we placed worms on a bacterial lawn and observed their location after a 24-h period. For the food choice assay, we set a plate with OP50 lawn on one side and *L. plantarum* lawn on the opposite side. The percentages of worms found on each bacterial lawn were monitored at specific time intervals to assess their food preference. The results showed worms tend to stay on OP50 as compared to the *L. plantarum* lawn (Fig. 1b and c), indicating *L. plantarum* is a less favorable food source. Interestingly, in the food choice assay, after OP50 was consumed on day 4, ~25% of worms switched to the *L. plantarum* lawn and consumed it (Fig. 1c), suggesting that worms can utilize *L. plantarum* food for growth if they previously ate OP50.

To further elucidate the stage at which developmental arrest occurred in the presence of *L. plantarum*, we assessed M lineage patterning as an indicator of worm development progression. In postembryonic development, a single mesodermal blast cell (M) undergoes division to produce a small number of additional mesodermal cells[11,12]. In hermaphrodites, the M divisions occurring in early larval development result in the formation of 14 striated body wall muscles, two sex myoblasts (SMs), and two coelomocytes[13]. By the L4 stage, the SMs undergo division, resulting in 16 SM descendants located near the vulval opening[11]. Here, we used *hlh-8::gfp* reporter expressed in the M-cell as the indicator. We observed that the GFP signal displayed the pattern characteristic of the stage between the late L1 stage to L2 stage in worms fed *L. plantarum* after 3 days, while worms fed OP50 exhibited a GFP signal pattern resembling the L4 stage (Fig. 1d). This observation strongly suggests that the development of worms fed with *L. plantarum* arrested or stalled at the late L1 stage to L2 stage.

### *L. plantarum*-induced development arrest is not due to bacterial avoidance

As *C. elegans* pharyngeal pumping filters bacteria and transports them from the stoma to the isthmus[14], we then asked if the size of *L. plantarum* (0.9–1.2 μm wide and 3–8 μm long[15]) is too large for L1 worms to feed. In order to see if *L. plantarum* can pass the worm pharynx and reach the gut, we used a green fluorescent dye carboxyfluorescein diacetate succinimidyl ester (CFSE) to stain *L. plantarum* and fed the fluorescent *L. plantarum* to *C. elegans*. After 2 h of incubation, we found L1 worms fed CFSE stained *L. plantarum* had green fluorescence signal both in the pharynx and the gut (Fig. 2a), indicating that the dietary effect from *L. plantarum* on worm development was not due to deficiency of bacteria uptake.

Since starvation causes animal physiological and behavioral changes, we wondered whether these changes are responsible for the developmental arrest in worms fed on *L. plantarum*. To test this hypothesis, newly isolated eggs were placed on the *L. plantarum* lawn to avoid starvation. However, the worms stayed at the L1/L2 stage 5 days after being placed on the *L. plantarum* lawn (Fig. 2b), suggesting the developmental arrest induced by *L. plantarum* is not because the worm avoided *L. plantarum* after hatching. To explore whether this developmental arrest effect is reversible, OP50 was added to worms exposed to *L. plantarum* for 6 days. After adding OP50, developmental arrested worms recovered to adults with viable progenies within 3 days (Fig. 2c). This suggests that the nutrient deficiency of *L. plantarum* induced a reversible developmental arrest to the host, which is a similar protective response to starvation. We next asked if *L. plantarum* is also a nutrient-deficient food source for another typical development arrested stage, dauer larvae. To test this, dauer larvae were isolated and fed either OP50 or *L. plantarum*. We found that dauer worms grew normally into adults when fed *L. plantarum* food (Fig. 2d). These findings suggest that *L. plantarum* lacks certain nutrients that are essential for the growth of early larval stage worms, but it provides sufficient nutrients for exit from the dauer stage to adulthood.

### A trace amount of *E. coli* can support the development of worms fed on *L. plantarum*

Previous studies have shown that when larvae are provided with inadequate food, larval growth can be restored by supplementing a trace amount (0.2 μl, $OD_{600} = 1$, $1.6 \times 10^8$ c.f.u.) of live *E. coli*[5]. To determine whether *E. coli* metabolites can compensate for some *L. plantarum* nutritional deficiency, L1 worms first were fed the same trace amount of live *E. coli* OP50 or BW25113. The optical refractivity of worms fed with 0.2 μl of bacterial culture changed so that they were easier to visualize, but they failed to develop substantial biomass (Fig. 3a).

Next, we fed the worms abundant *L. plantarum* together with trace live *E. coli*. We found that larval growth was recovered with the supplemented *E. coli* (Fig. 3b), and the progeny from those worms were also able to grow on *L. plantarum*, suggesting that factors from the trace amount of *E. coli* stimulate *C. elegans* surpass L1/L2 stage so they can use *L. plantarum* as nutrients. To characterize further the active compounds from *E. coli*, we supplemented *L. plantarum* with heat-treated, UV-treated *E. coli* or *E. coli*-conditioned media. We observed that while live *E. coli* promoted *C. elegans* growth of worms on *L. plantarum*, none of the above conditions were as effective (Fig. 3c). Thus, *E. coli* factors that rendered the *L. plantarum* edible are not secreted, heat nor UV stable. Increasing evidence suggests that various bioactive dietary factors may modify the epigenome and could thus be incorporated into an epigenetic diet[16]. Here we sought to determine whether a diet supplemented with *L. plantarum* could render the diet edible for their offspring. To test this, gravid worms were fed trace amounts of *E. coli* with an abundant supply of *L. plantarum*, then bleached to obtain L1 worms, which were subsequently placed on an *L. plantarum* lawn. As a control, L1 worms from mothers fed only *E. coli* were used. After 5 days on the plate, the worms remained in the early developmental stage, similar to the control group (Fig. 3d). These results suggest that potential epigenetic modifications resulting from the *L. plantarum* diet were insufficient to render the diet edible for their offspring.

### A high-throughput screen of *E. coli* single gene deletions identifies the genes required for *C. elegans* to fed and develop on *L. plantarum*

To identify systematically the bacterial components that act as nutrients to recover worm development, we performed a genome-wide screen using the Keio collection library. This library contains 3985 single gene knockout *E. coli* strains, with individual genes replaced with a kanamycin cassette in the K12 BW25113 background[17]. To conduct the screen, ~100 synchronized L1 worms were placed onto the plates with a trace amount of *E. coli* mutants from the Keio collection, together with *L. plantarum* (Fig. 4a). We then monitored the worms' development for 4–6 days. The wild-type parental strain *E. coli* BW25113 was used as the positive control. In the primary screen, we selected *E. coli* candidates that potentially delayed worm development. We performed a secondary screen with biological triplicates to

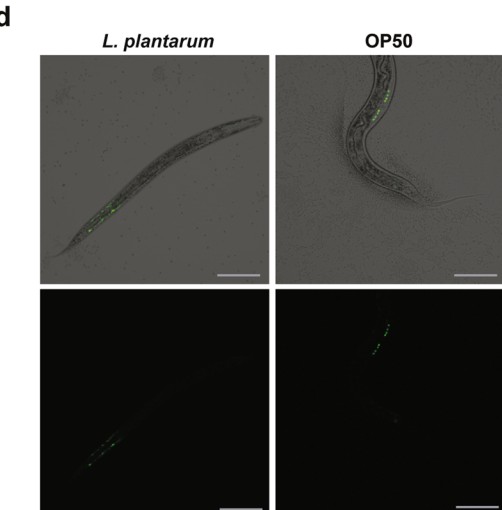

**Fig. 1 | *L. plantarum* induces early-stage larval arrest in *C. elegans*. a** Microscope image and bar graph showing that worms fed with *L. plantarum* were arrested at the early larvae stage three days after hatching. Scale bar, 1000 µm. **b** Schematic drawing and quantitative data of the food dwelling assay. Circles indicate the bacterial lawn of OP50 (upper) and *L. plantarum* (bottom), respectively. The animals were scored 24 h after L1 worms were placed on the food spot. **c** Schematic drawing, microscope images and quantitative data of the food choice assay. Synchronized L1 worms were placed in the center spot (origin). OP50 (upper) and *L. plantarum* (bottom) bacteria were placed on opposite sides of the plate. The percentages of worms on each bacterial lawn were calculated at the indicated time. Worms moved to *L. plantarum* lawn and ate it after 4 days when OP50 was totally consumed. Scale bar, 1000 µm. **d** Synchronized L1 worms with *hlh-8::gfp* reporter expressed in the M-cell were fed with either *L. plantarum* or OP50 for 3 days. Fluorescent microscope images showing the GFP signal distribution. Scale bar, 50 µm. All data are representative of at least three independent experiments. *n* = number of worms scored. Data are represented as mean ± SEM. Significance determined by unpaired two-tailed Student's *t*-test. * indicates *P*-value < 0.05, ** indicates *P*-value < 0.01, N.S. indicates non-significant difference.

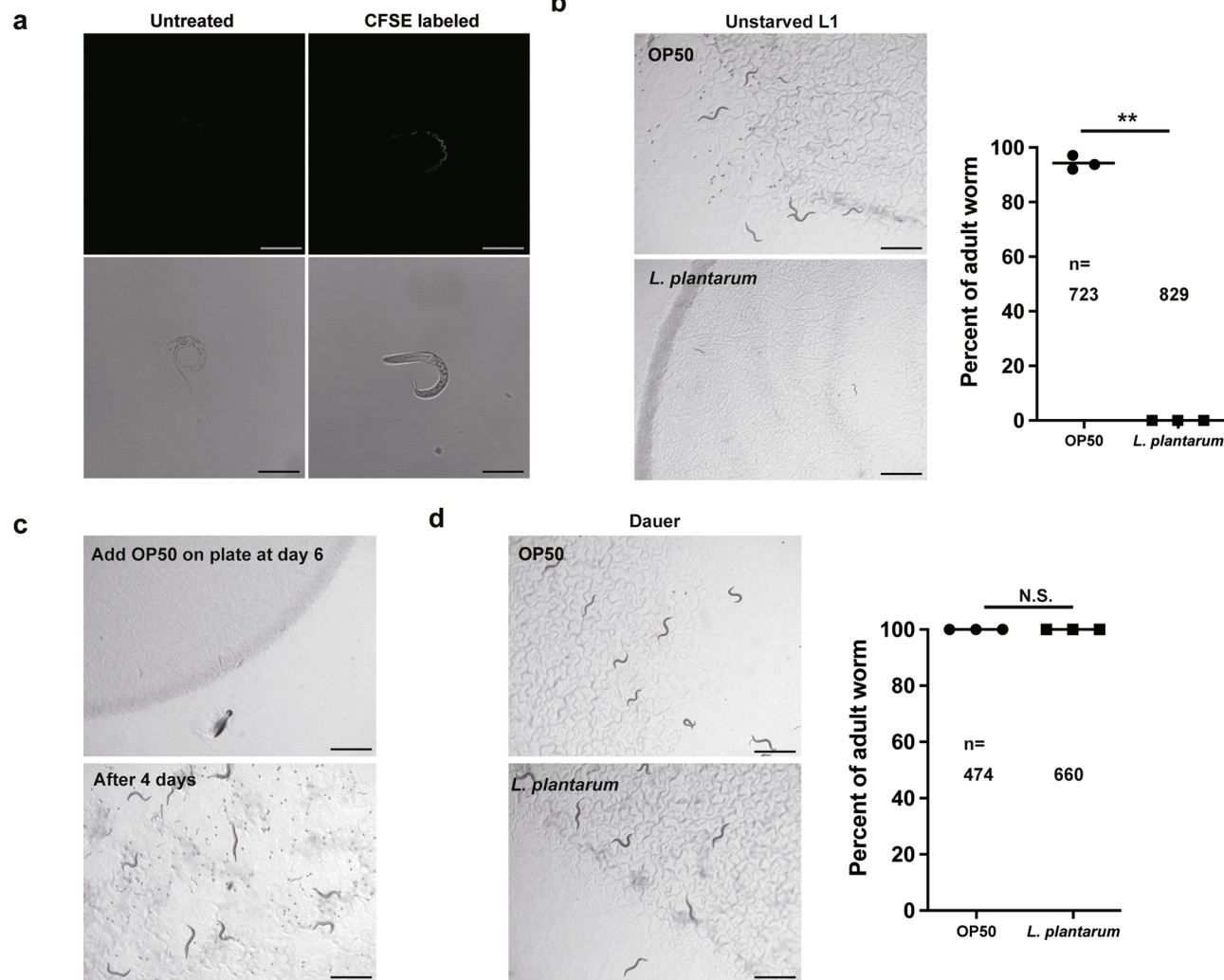

**Fig. 2 | *L. plantarum*-induced development arrest is not due to bacterial avoidance. a** Microscope images showing the distribution of GFP signal in the newly hatched L1 worms fed either unstained *L. plantarum* or CFSE-labeled fluorescent *L. plantarum*, 2 h after feeding. Scale bar, 50 μm. **b** Microscope images and bar graph demonstrating that newly hatched worms fed on *L. plantarum* without starvation were arrested at the early larval stage three days after being placed on the bacteria. Scale bar, 1000 μm. **c** Worms grown on *L. plantarum* for 6 days can recover to adults with viable progeny 4 days after switching to OP50 food. Scale bar, 1000 μm.

**d** Microscope images and bar graph showing that dauer larvae fed *L. plantarum* had a similar percentage of adult worms to those fed on OP50, 2 days after being placed on the bacteria. Scale bar, 1000 μm. All data are representative of at least three independent experiments. *n* = number of worms scored. Data are represented as mean ± SEM. Significance determined by unpaired two-tailed Student's *t*-test. * Indicates *P*-value < 0.05, ** indicates *P*-value < 0.01, N.S. indicates non-significant difference.

narrow the candidates to 421 *E. coli* mutants (Fig. 4a). After a third screening, we confirmed 29 mutant *E. coli* strains that failed to support worm growth (Fig. 4b, Table 1).

### *E. coli* vitamin B6 is essential for post-embryonic development in *C. elegans*

To classify the function of these 29 genes essential for recovering *C. elegans* early larval stage development, we performed a KEGG pathway enrichment analysis. We found that the genes are involved in a variety of metabolic pathways, including vitamin B6 metabolism, ferric enterobactin transports, biosynthesis of siderophore group non-ribosomal peptides, cysteine and methionine metabolism, pantothenate and CoA biosynthesis, and some unclassified pathways (Table 1). Out of these 29 genes, 5 of them are involved in the *E. coli* vitamin B6 biosynthesis pathway (Table 1). In contrast to eukaryotes, *E. coli* is capable of synthesizing the active form of vitamin B6, pyridoxal 5′-phosphate (PLP), through both de novo and salvage biosynthesis pathways (Fig. 5a). In order to investigate the role of

these two pathways in regulating the growth of worms fed with *L. plantarum*, we fed the worms with various *E. coli* mutants and evaluated their growth by measuring worm size. Our results indicate that the development of worms arrested at early stage when fed with a trace amount of *E. coli* mutants with disrupted vitamin B6 de novo biosynthesis (Fig. 5b). This suggests that *E. coli* vitamin B6 de novo biosynthesis pathway plays an important role in recovering early-stage larval growth in the context of *L. plantarum* diet. To eliminate the possibility of contamination of these *E. coli* mutant strains, we performed PCR using gene-specific primers and verified the correct gene deletion in the knockout mutants (Supplementary Fig. 1).

To further investigate the *E. coli* metabolites involved in vitamin B6 metabolism-regulating arrested larvae development, the intermediate pyridoxine (PN) and final metabolite pyridoxal 5′-phosphate (PLP) were supplemented to the worms fed on trace *E. coli pdxJ* or *E. coli pdxH* mutants together with plenty of *L. plantarum*. We found the worm development completely recovered by dietary supplementation of the *pdxH* downstream metabolite PLP (Fig. 5c, Supplementary Fig. 2). Meanwhile, supplementing

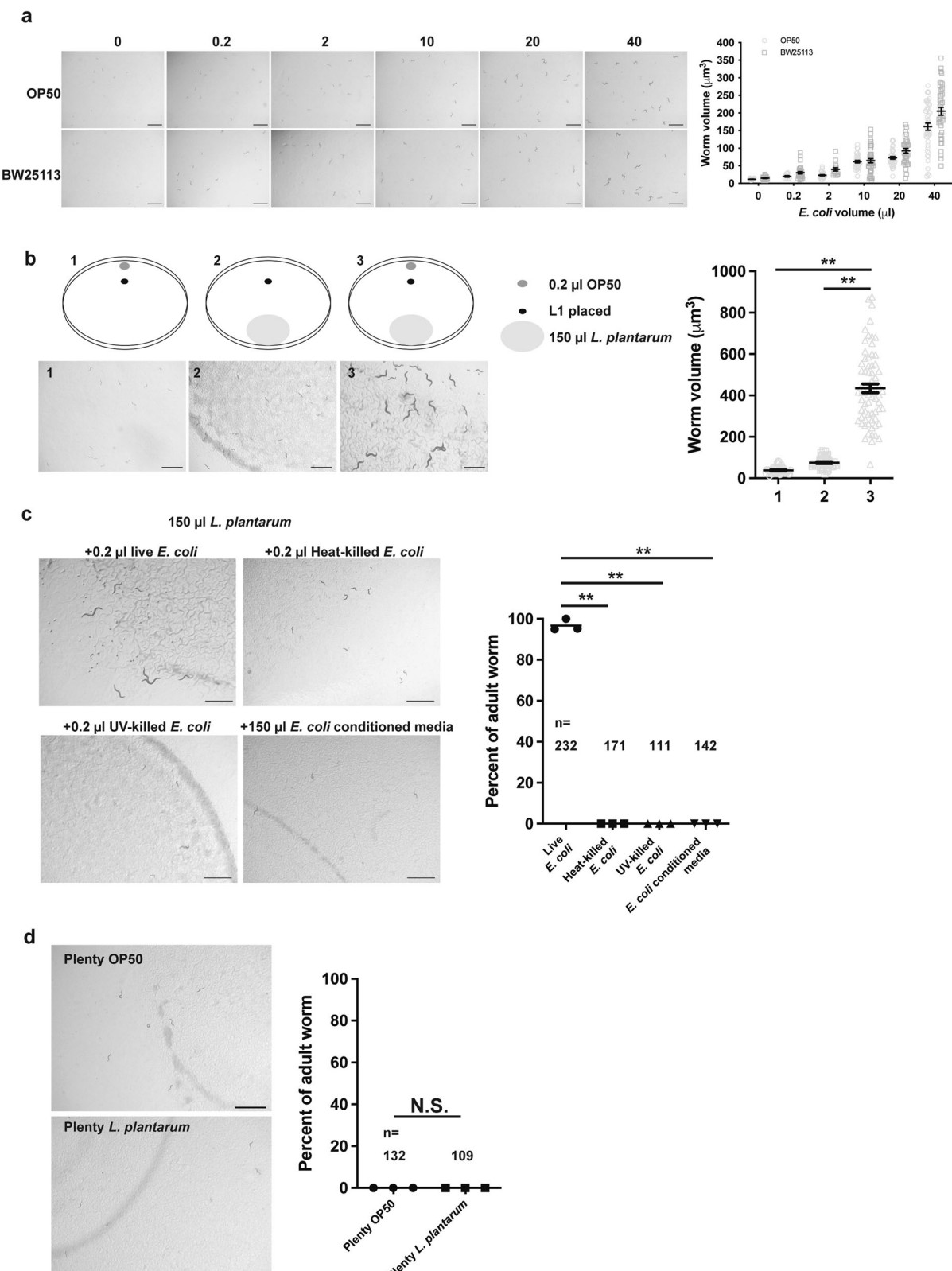

with intermediate PN rescued worm growth on *pdxJ* mutant but not *pdxH* mutant (Fig. 5d, Supplementary Fig. 2), indicating that the final product PLP is essential for *C. elegans* larval development. To determine whether the growth effect is restricted to a specific bacterial background, we added PLP only, PLP supplemented with the cofactor Nicotinamide, and PLP with

trace amounts of UV-killed *E. coli pdxH* mutant to the worms that were fed on *L. plantarum*, individually. We found that supplementing 1 mM PLP without live *pdxH* mutant did not promote the worm growth (Fig. 5e), suggesting that the beneficial role of PLP for larval growth either requires some live *E. coli* molecules to deliver the PLP to *C. elegans*, or *C. elegans* is

**Fig. 3 | A trace amount of *E. coli* can support the development of worms fed on *L. plantarum*. a** Microscope image and bar graph showing that synchronized L1 worms fed on inadequate amounts of *E. coli* OP50 or BW25113 alone remained small. The volume of the worms was measured 4 days after the larvae were placed on the plates. Number of worms scored in the OP50 diet is 42, 32, 46, 46, 46, 46; Number of worms scored in the BW25113 diet are 46, 46, 40, 46, 46, 46. **b** Schematic drawing, microscope images, and bar graphs showing that a trace amount of *E. coli* OP50 supports the postembryonic growth of worms fed *L. plantarum*. The volume of the worms was measured 4 days after the larvae were placed on the plates. **c** Representative images and bar graph showing synchronized worms fed on 150 µl *L.*

*plantarum* together with 0.2 µl live *E. coli*, 0.2 µl heat-killed *E. coli*, 0.2 µl UV-killed *E. coli*, or 150 µl *E. coli* conditioned media for 4 days. **d** L1 worms from mothers fed only *E. coli* or from mothers fed trace amounts of *E. coli* with an abundant supply of *L. plantarum* were placed on a *L. plantarum* lawn, respectively. Worm growth was measured 5 days after larvae were placed on the plates. All data are representative of at least three independent experiments. Scale bar, 1000 µm. *n* = number of worms scored. Data are represented as mean ± SEM. Significance determined by unpaired two-tailed Student's *t*-test. * Indicates *P*-value < 0.05, ** indicates *P*-value < 0.01, N.S. indicates non-significant difference.

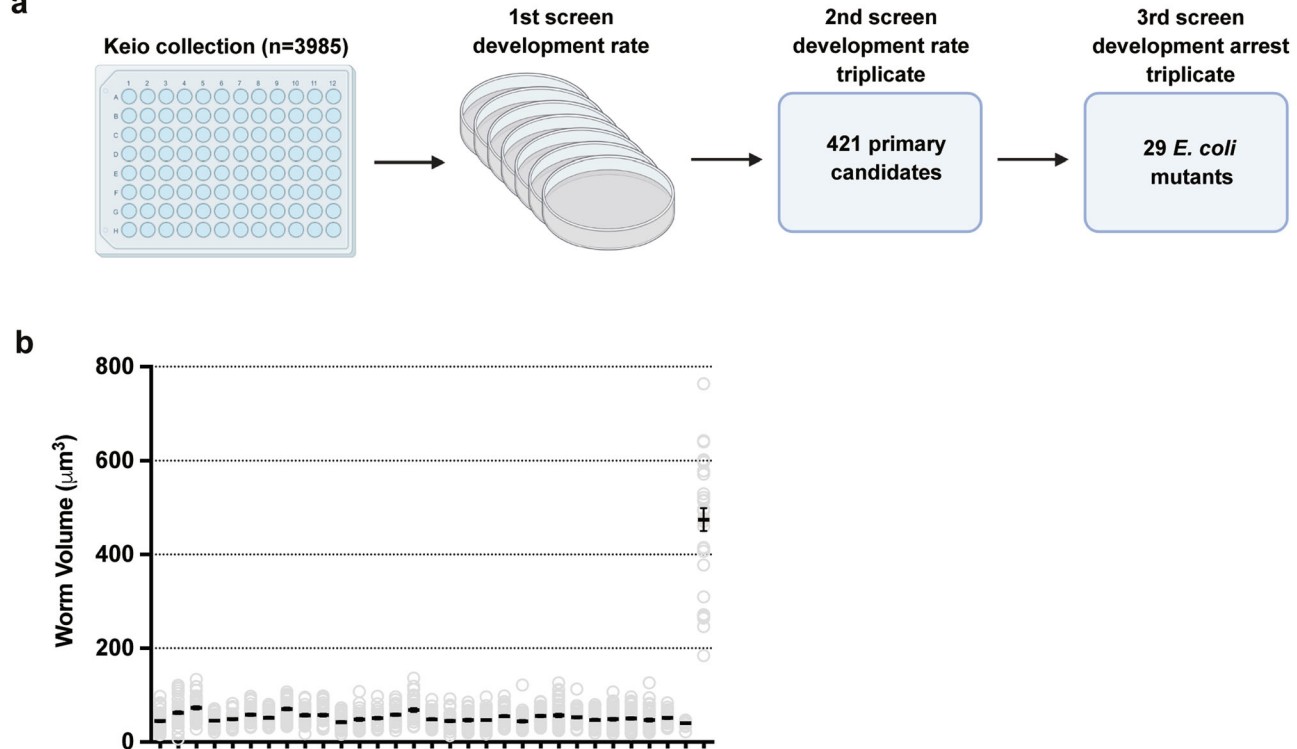

**Fig. 4 | High-throughput screen for *E. coli* single gene deletions affecting *C. elegans* development when fed with *L. plantarum*. a** Schematic of the genome-wide screens. **b** Bar graph showing the impact of *E. coli* mutant strains from the library on the development of worms fed on 150 µl *L. plantarum*. The volume of the worms was measured 4 days after synchronized L1 worms were placed on the plates. Number of worms scored in each group are 49, 54, 46, 37, 36, 36, 31, 42, 30, 40, 43, 30, 31, 47, 39, 38, 35, 35, 53, 56, 39, 35, 46, 41, 42, 45, 43, 41, 30, 18, 31. Data are represented as mean ± SEM.

---

dependent on some downstream bacterial usage of PLP. As trace amounts of *E. coli pdxH* mutant cannot support the growth of worms fed on *L. plantarum*, we investigated whether a plentiful amount of the *pdxH* mutant alone could support worm development. We placed worms on an *E. coli pdxH* mutant lawn and used BW25113 lawn as the control. The results show that the lack of *pdxH* does not affect worm growth (Fig. 5f). Next, we explored whether other components from *E. coli* could compensate for the function of PLP and support the growth of worms fed on *L. plantarum*. To investigate this, we supplemented worms with an *L. plantarum* diet with varying concentrations of the *pdxH* mutant. We found that supplementation of no less than 5 µl of *pdxH* mutant (OD600 = 1) could partially recover the growth of the worms (Fig. 5g). This suggests that the function of *pdxH* can be compensated for by increasing the amount of the *E. coli* mutant. The *pdxH* mutant was grown in rich nutrient-containing yeast extract, which provided the vitamins required for the mutant's enzyme functions. Taken together, these results demonstrate that *E. coli pdxH* mutant provides limited nutrition, with vitamin B6 deriving from its prior culturing conditions, supporting worm growth.

### Bacterial *pdxH* affects host development by coordinating host metabolic processes and PLP binding activity

To investigate the signaling mechanism that leads to the developmental arrest of *C. elegans* fed on *L. plantarum* due to the absence of *pdxH* in *E. coli*, we carried out RNA sequencing and expression analysis to identify the host pathways regulated by *E. coli pdxH* activity. Specifically, we compared the RNA-seq results of worms fed an *L. plantarum* diet with either a trace amount of *E. coli pdxH* mutant or wild-type *E. coli* BW25113. We identified 120 genes with increased expression and 830 genes with reduced expression with a Gfold (0.01) value (which indicates the fold change in gene expression) over 0.5 (Supplementary Data 1), suggesting widespread effects of the lack of *pdxH* in *E. coli* diet on host gene expression in *C. elegans*. Gene ontology (GO) analysis showed that upregulated genes were mainly involved in the metabolic process and oxidation-reduction process (Fig. 6a), while downregulated genes were mainly involved in development and growth (Fig. 6b), consistent with the developmental arrest phenotype. To gain insight into the crosstalk of vitamin B6 biosynthesis in *E. coli* with host vitamin B6 metabolism, we assessed the expression level of genes involved in

**Table 1 | Gene annotation of 29 identified *E. coli* genes from the screening**

| KEGG pathway | Gene | Description |
|---|---|---|
| Vitamin B6 Metabolism | *serC* | Phosphoserine aminotransferase |
| | *pdxJ* | Pyridoxine 5'-phosphate synthase |
| | *pdxB* | Erythronate-4-phosphate dehydrogenase |
| | *pdxA* | 4-hydroxythreonine-4-phosphate dehydrogenase |
| | *pdxH* | Pyridoxine/pyridoxamine 5'-phosphate oxidase |
| Ferric enterobactin transporters | *Fes* | Enterochelin esterase |
| | *fepB* | Ferric enterobactin-binding periplasmic protein |
| | *fepC* | Ferric enterobactin transport ATP-binding protein |
| | *fepG* | Ferric enterobactin transport system permease protein |
| | *fepD* | Ferric enterobactin transport system permease protein |
| Biosynthesis of siderophore group nonribosomal peptides | *entF* | Enterobactin synthase component F |
| | *entA* | 2,3-dihydro-2,3-dihydroxybenzoate dehydrogenase |
| | *entB* | Enterobactin synthase component B |
| Cysteine and methionine metabolism | *cysE* | Serine acetyltransferase |
| | *cysK* | Cysteine synthase A |
| Pantothenate and CoA biosynthesis | *panB* | 3-methyl-2-oxobutanoate hydroxymethyltransferase |
| | *panE* | 2-dehydropantoate 2-reductase |
| | *panC* | Pantothenate synthetase |
| Unclassified | *hns* | DNA-binding transcriptional dual regulator H-NS |
| | *rpoN* | RNA polymerase, sigma 54 (sigma N) factor |
| | *yaiS* | putative deacetylase |
| | *mdtH* | multidrug efflux pump |
| | *ybhR* | putative ABC exporter membrane subunit |
| | *guaA* | GMP synthetase |
| | *cysQ* | 3'(2'),5'-bisphosphate nucleotidase |
| | *bioA* | Adenosylmethionine-8-amino-7-oxono-nanoate aminotransferase |
| | *bioH* | Pimeloyl-[acyl-carrier protein] methyl ester esterase |
| | *aroA* | 3-phosphoshikimate 1-carboxyvinyltransferase |
| | *aroE* | Shikimate dehydrogenase (NADP+) |

host vitamin B6 metabolism. The RNA-seq results showed fluctuated gene transcription in host PLP binding (Fig. 6c). Further qPCR analysis confirmed the relative mRNA level changes of two representative genes *eppl-1* and *F26H9.5*, compared to the internal reference gene *act-1* (Fig. 6d). These results suggest a fundamental role of *E. coli* vitamin B6 in regulating host PLP binding activity and restoring worm development with a diet of *L. plantarum*.

### The developmental arrest induced by the *L. plantarum* diet in worms does not rely on FoxO/DAF-16 activation

The insulin-like signaling (IIS) pathway plays a pivotal role in regulating developmental arrest and gene expression in *C. elegans* when confronted with adverse conditions such as nutrient depletion and oxidative stress[18,19]. The FOXO transcription factor DAF-16, a downstream effector of the IIS

pathway, is responsible for integrating stress signals and translocating from the cytoplasm to the nucleus[20,21]. To shed light on the participation of DAF-16 in worm developmental arrest mediated by *L. plantarum*, we investigated the subcellular localization of DAF-16 in worms exposed to various conditions, including those exclusively fed *L. plantarum*, *L. plantarum* in combination with 0.2 μl of BW25113, *L. plantarum* with 0.2 μl of the *E. coli pdxH* mutant, and *L. plantarum* with 0.2 μl of the *E. coli pdxH* mutant in addition to 1 mM PLP. We found that worms fed *L. plantarum* only or *L. plantarum* with 0.2 μl of the *E. coli pdxH* mutant exhibited a significant increase in the nuclear localization of DAF-16 after 4 days incubation when compared to worms fed *L. plantarum* with 0.2 μl of wild-type *E. coli* BW25113 (Fig. 6e). In addition, PLP supplementation reversed the DAF-16 nuclear translocation (Fig. 6e), indicating the nuclear translocation of DAF-16 in worms fed *L. plantarum* with 0.2 μl of the *E. coli pdxH* mutant is likely due to stress associated with deficiency in PLP or a decrease in a downstream PLP-regulated process. To further clarify the role of *daf-16*, we asked whether *daf-16(mu86)* null mutants developmentally arrest or can develop on *L. plantarum*. When this mutant was exclusively fed *L. plantarum*, the worms exhibited developmental arrest (Fig. 6f). This suggests that processes that lead to developmental arrest do not rely on *daf-16* for activation.

### Discussion

Animals require essential nutrients to support their development. Microbes, either through commensal interactions or as food sources, provide certain essential nutrients that cannot be synthesized by the animal. Our study shows that *L. plantarum*, when provided as a nutrient-deficient food source for *C. elegans*, can induce reversible developmental arrest. This finding enhances our understanding that larvae hatching in the presence of food but lacking essential nutrients fail to develop. Notably, a trace amount of *E. coli* enables worms to overcome this developmental arrest and complements the nutrient deficiency in *L. plantarum*. With this *L. plantarum*-induced developmental arrest model, we screened an *E. coli* single deletion library and found that the alteration of bacterial genetic composition influences *C. elegans* recovery from developmental arrest. As a result, the bacterial *pdxH* is not nutritionally adequate for the development of worms fed on *L. plantarum*. However, the supplementation of the *pdxH* downstream metabolite PLP compensates for the nutrient deficiency and recovers developmental arrested worm growth. These results emphasized the importance of coexisting *E. coli* acts synergistically with *L. plantarum* and provides essential nutrients to larval development in *C. elegans*.

Interestingly, our RNA-seq analysis revealed multiple PLP binding genes are decreased in larval *C. elegans* fed with *L. plantarum* (Fig. 6c), one of which is *tatn-1* (Fig. 6c). *tatn-1* gene encodes a tyrosine aminotransferase (TAT), previous studies have shown that TAT is a PLP-dependent enzyme catalyzes the conversion of tyrosine to 4-hydroxyphenylpyruvate, which is a rate-limiting step to metabolize tyrosine to fumarate and acetoacetate[22,23]. RNAi knockdown of *tatn-1* gene expression can stall larval development[24,25]. Here, we hypothesize that the PLP from *E. coli* functions as a coenzyme of worm TAT, the absence of PLP from the dietary bacteria leads to a reduced expression of *tatn-1* (Fig. 6c), subsequently leading to downstream deficiency in fumarate and acetoacetate or detrimental buildup of tyrosine and the developmental arrest. It is possible that *E. coli* is responsible for delivering vitamin B6 to *C. elegans* TATN-1, either through a bacterial conjugate (such as a PLP-binding bacterial protein) or it is equally possible that *C. elegans* does not directly require vitamin B6, but instead, it relies on specific rate-limiting molecules from *E. coli* that require vitamin B6 for their synthesis (Fig. 6g).

In most animals, including *C elegans*, the PLP biosynthetic pathway is dysfunctional, requiring them to obtain vitamin B6 from their diet or microbiome[26]. Some species of microbiota including Lactobacillus lack the ability to biosynthesize vitamin B6[27], which makes synergistic in providing nutrients to the host important between microbes. Colonization of bacteria that has genomic diversity in vitamin B6 synthesis in *C. elegans* is beneficial to the health and life traits of the host[28]. In a chemically defined medium, *C. elegans* maintenance medium (CeMM), the active form of bacterial vitamin

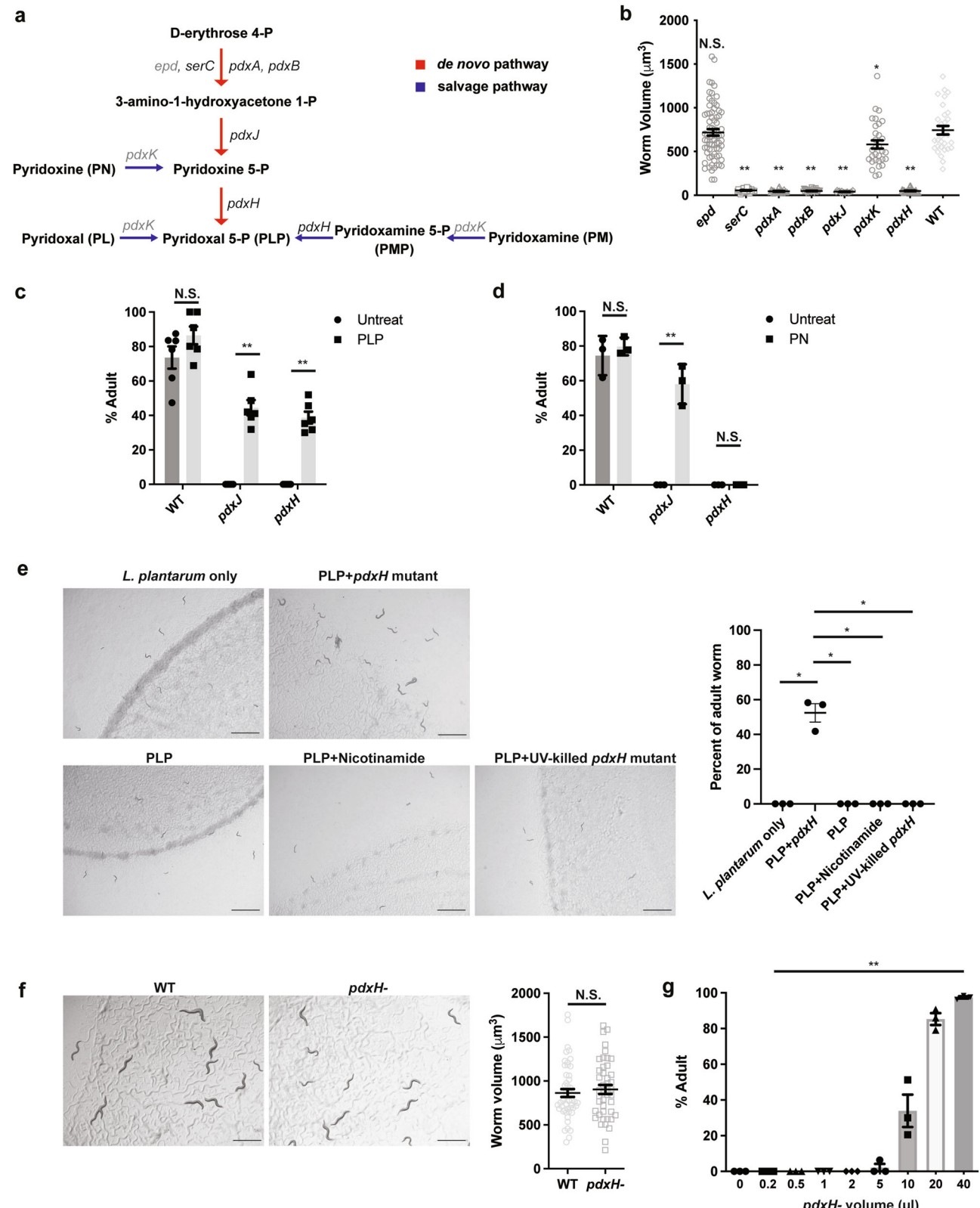

B6, PLP, is indispensable for worm population growth and health[29]. Lack of autonomous *E. coli* synthesis of PLP does not affect worm growth as the diet alone, provided sufficient live mutant bacteria is fed to the worms (Fig. 5f). This observation suggests that PLP-binding enzymes in living *pdxH* mutants can either (i) serve as a delivery vehicle for the vitamin to *C. elegans* enzymes, such as tyrosine aminotransferase, or (ii) vitamin B6 might

be required for *E. coli* PLP-dependent enzymes to synthesize essential molecules, such as amino acids, that might be deficient in the context of a *L. plantarum* diet for larval *C. elegans*[30]. When *pdxH* mutant is cultivated in nutritionally rich LB broth (usually containing yeast extract), it likely provides vitamin B6 along with all of the amino acids that are synthesized by vitamin B6 binding enzymes. It's possible that the essential enzymes in the

**Fig. 5 | E. coli vitamin B6 is essential for post-embryonic development in C. elegans. a** The vitamin B6 synthesis pathway in *E. coli* with genes essential for worm growth from the screening labeled in black. **b** Bar graph showing the development of synchronized L1 worms fed 0.2 μl *E. coli* mutants with disrupted vitamin B6 biosynthesis together with 150 μl *L. plantarum*. The volume of the worms was measured 3 days after the larvae were placed on the plates. Number of worms scored in each group are 72, 31, 25, 26, 16, 31, 37, 30. **c** The final metabolite pyridoxal 5′-phosphate (PLP) was supplemented to worms fed on 0.2 μl *E. coli pdxJ* or *E. coli pdxH* mutants together with 150 μl *L. plantarum*. The volume of the worms was measured 3 days after the larvae were placed on the plates. **d** The intermediate pyridoxine (PN) was supplemented to the worms fed on 0.2 μl *E. coli pdxJ* or *E. coli pdxH* mutants together with 150 μl *L. plantarum*. The volume of the worms was measured 3 days after the larvae were placed on the plates. **e** Worms were fed on *L. plantarum* together with PLP only, PLP supplemented with the cofactor Nicotinamide, and PLP with trace amounts of UV-killed *E. coli pdxH* mutant, respectively. **f** Worms were placed on an *E. coli pdxH* mutant lawn and a BW25113 lawn as the control. Worm growth was measured 2 days after L1 was placed on the plates. Number of worms scored in each group are 50 and 44. **g** Worms were fed with an *L. plantarum* diet with varying concentrations of the *pdxH* mutant. Worm growth was measured 4 days after L1 was placed on the plates. All data are representative of at least three independent experiments. Scale bar, 1000 μm. *n* = number of worms scored. Data are represented as mean ± SEM. Significance determined by unpaired two-tailed Student's *t*-test. * Indicates *P*-value < 0.05, ** indicates *P*-value < 0.01, N.S. indicates non-significant difference.

*pdxH* mutant diet are loaded with vitamin B6 from its yeast extract containing growth media. Consequently, the greater the quantity of *pdxH* mutant added to *C. elegans*, the more indirectly yeast-derived vitamin B6 is introduced, thereby supporting worm growth.

The ability of organisms to sense their nutritional environment enables them to alter their growth and metabolism accordingly. Previous studies on larval arrest have shown that the insulin/insulin-like growth factor signaling (IIS) pathway plays a crucial role in sensing the nutritional environment and regulating entry into arrest[31]. The IIS pathway is the principal regulator linking nutrient levels to metabolism and development in *C. elegans*. DAF-2, which is an ortholog of the insulin/IGF-1 transmembrane receptor, plays a key role in regulating worm development[32]. DAF-16 is a widely expressed transcription factor belonging to the Forkhead family, it regulates the expression of various genes involved in development, longevity, and stress resistance[32]. DAF-2 functions primarily through the regulation of DAF-16 to control worm developmental arrest[33]. In this study, although DAF-16 does translocate into the nucleus upon *L. plantarum* feeding, it does not seem to be the primary cause of developmental arrest or stalling. The developmental defect from the *L. plantarum* diet is possibly due to a lack of rate-limiting factors crucial for robust developmental progression. DAF-16 might just promote a coping mechanism to sustain worm survival until an adequate amount of rate-limiting substance is acquired. This rate-limiting substance could potentially be vitamin B6 delivered via a bacteria-derived vehicle or other *E. coli* factors that rely on vitamin B6 for their synthesis. As we gain a deeper understanding of the regulatory network controlling post-embryonic development, we may discover additional signals and signaling centers in the future. Given that *Lactiplantibacillus* is also a commensal bacterium in many other metazoans, including humans, investigating the interaction between *Lactiplantibacillus* and *C. elegans* may offer new insights into evolutionarily conserved probiotic effects.

## Methods
### Nematode and bacterial strains
Wild type N2 Bristol, PD4667 *hlh-8::gfp + dpy-20*(+), TJ357 *daf-16p::daf-16a/b::gfp + rol-6(su1006)*, and CF1037 *daf-16(mu86)* I was obtained from the Caenorhabditis Genetics Center (CGC) and maintained on nematode growth medium (NGM) 6 cm Petri dish plates at room temperature. All experiments were performed with synchronized L1 stage hermaphrodite animals. *E. coli* OP50 and the Keio collection parent strain *E. coli* BW25113 were grown at 37 °C in Lysogeny broth (LB) Miller's version. *E. coli* deletion strains were grown at 37 °C in LB with 50 μg/ml kanamycin. *Lactiplantibacillus plantarum* ATCC8014 was grown at 37 °C in Lactiplantibacillus MRS broth for 24 h. Bacterial cultures were seeded on NGM plates and dried for 1 h at room temperature prior to use in experiments.

### CFSE staining
The green fluorescent dye 5(6)-Carboxyfluorescein diacetate succinimidyl ester (CFSE, MedChemExpress) was used to track the bacteria. 1 ml of *L. plantarum* cultured 24 h was harvested and centrifuged at $4000 \times g$ for 5 min. The resulting pellet was resuspended in 1 ml of 10 μM CFSE in PBS and incubated in the shaker at 37 °C at 200 rpm for 30 min while avoiding

light. After staining, the bacteria were washed three times with PBS and resuspended in PBS.

### Dauer larvae isolation
The N2 hermaphrodites were grown with OP50 at room temperature for ~10 days, during which there was an abundance of dauer larvae on the plate. The worms were then washed off the plate, and the worm pellet was resuspended in 1% SDS (sodium dodecyl sulfate). The worms were incubated in 1% SDS for 30 min with gentle agitation. Then, the worms were washed 1–5 times with M9 buffer to remove all SDS. To remove the carcasses of non-dauer larvae, 6 ml of 30% ice-cold sucrose was added to the worm pellet and then centrifuged at 4 °C for 5 min at a speed of 3000 rpm. The floating worms were aspirated, and the pellet was washed twice with M9 buffer to remove excess sucrose. To eliminate the potential existing OP50 in the gut, the isolated dauer worms were washed twice with 100 μg/ml gentamycin, each wash lasting 10 min. After washing three times with M9 to remove gentamycin, the worms were suspended in the M9 buffer.

### E. coli Keio collection screen
The MRS broth-cultured *L. plantarum* was concentrated to OD600 = 10 and resuspended in an M9 buffer. Subsequently, 150 μl of *L. plantarum* suspension ($1.2 \times 10^8$ c.f.u.) was added to one side of the peptone-free NGM plate. The *E. coli* mutants were cultured overnight at 37 °C in LB medium with 10 mg/ml kanamycin. Then, 0.2 μl of each mutant culture (OD600 = 1, $1.6 \times 10^5$ c.f.u.) was seeded to the other side of *L. plantarum* plate. Approximately 300 synchronized L1 worms were added to the screen plate and placed at room temperature. Then worm sizes were scored on days 3 and 4. Each mutant in the library was screened once for the primary screen. For the secondary screening, 421 candidate mutants were screened in triplicate to confirm the slow-growth phenotype.

### PCR verification of deletions
Single gene deletion *E. coli* strains were streaked onto LB agar plates containing 50 μg/ml kanamycin. Subsequently, PCR was performed on individual colonies using genomic and kanamycin-cassette-specific primers. Genomic primers were designed specifically for each strain at a distance of 100–300 bases upstream of the start codon of the gene, and 100–300 bases downstream of the stop codon. The primers used are listed in Supplementary Table 1. The PCR products were then analyzed for the correct size by agarose gel electrophoresis.

### Metabolites supplementation
Pyridoxine (PN, Fisher BioReagents, 1 mM) and pyridoxal 5′-phosphate (PLP, Thermo Scientific, 1 mM) were dissolved in water and added to *E. coli* mutants and then spotted onto the plates.

### Total RNA extraction
*C. elegans* at the same stage grown on *L. plantarum* together with a trace amount (0.2 μl, $1.6 \times 10^5$ c.f.u.) of wild-type BW25113 or *pdxH* mutant were collected from peptone-free NGM plates and washed three times with M9 buffer. Then 250 μl of TRIzol reagent (Invitrogen) was added to the worm

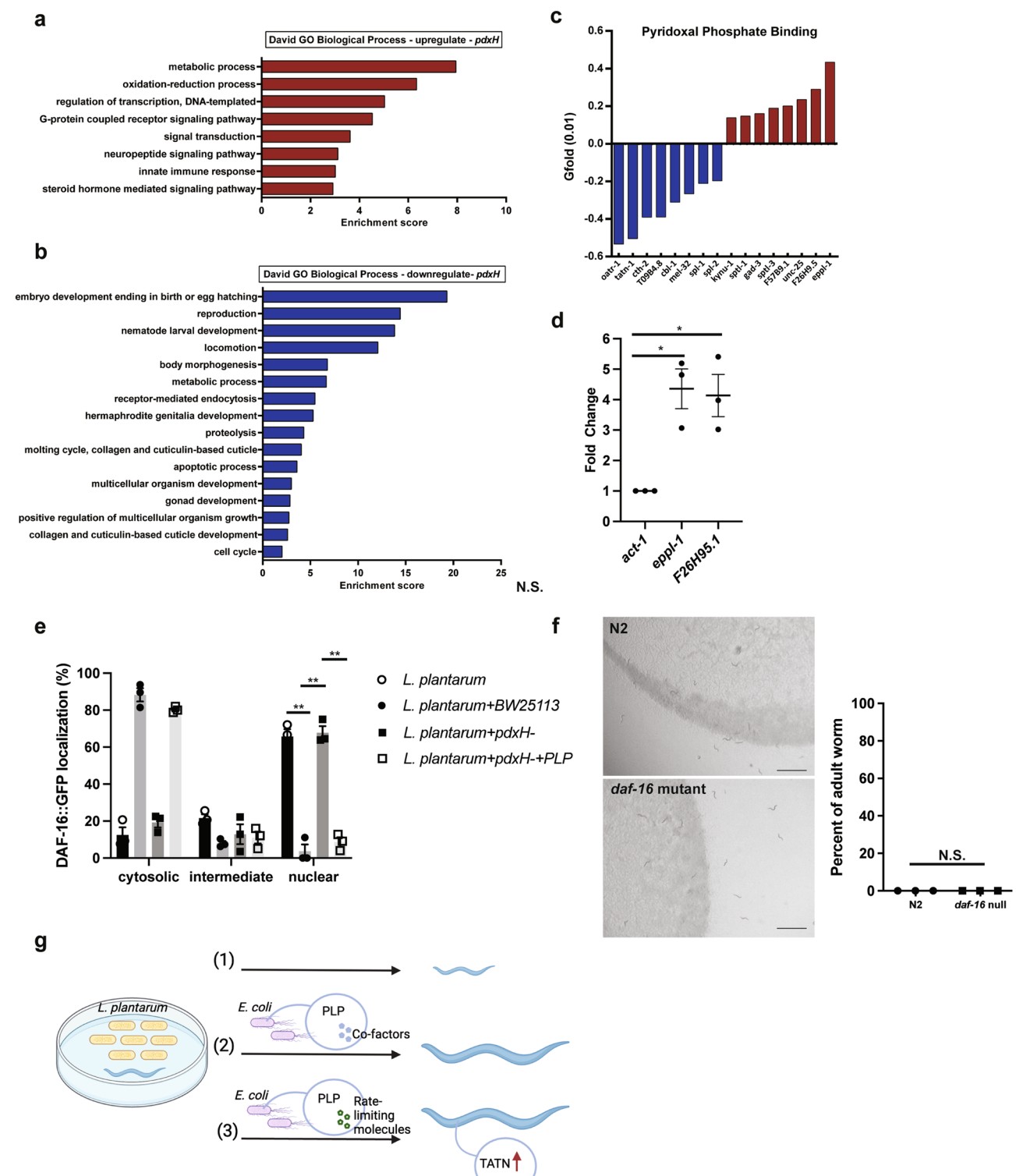

**Fig. 6 | The function of *E. coli pdxH* relies on its ability to coordinate host metabolic processes and PLP-binding activity. a** The most enriched gene ontology (GO) terms in upregulated genes in worms fed on *E. coli pdxH* vs WT *E. coli* together with 150 µl *L. plantarum*. **b** The most enriched gene ontology (GO) terms in downregulated genes in worms fed on 0.2 µl *E. coli pdxH* vs. WT *E. coli* together with 150 µl *L. plantarum*. **c** RNA-seq results reveal the expression levels of genes involved in host vitamin B6 metabolism. Down-regulated genes are indicated in blue, and up-regulated genes are indicated in red. **d** qPCR analysis shows the relative changes in the mRNA level of two representative genes, *eppl-1* and *F26H9.5*, compared to the internal reference gene *act-1*. **e** Bar graph demonstrating the activation of DAF-

16::GFP nuclear translocation in worms fed *L. plantarum* only or *L. plantarum* together with 0.2 µl *E. coli pdxH* compared to the worms fed *L. plantarum* together with 0.2 µl *E. coli pdxH* or wt *E. coli* BW25113. **f** Representative images and bar graphs showing *daf-16* null worms were fed either *L. plantarum* or OP50. Scale bar, 1000 µm. Data are presented as mean ± SEM in three independent experiments; Significance is determined by unpaired two-tailed Student's *t*-test. * Indicates *P*-value < 0.05, ** indicates *P*-value < 0.01, N.S. indicates non-significant difference. **g** A proposed model suggests that bacterial PLP acts as a cofactor of bacterial-derived molecules, promoting *L. plantarum* consumption and worm growth.

pellet and worms were homogenized for 5 min. RNA was isolated by adding 50 μl of chloroform, followed by centrifugation to separate the aqueous phase. The aqueous phase was transferred to a new tube and mixed with isopropanol (125 μl) to precipitate RNA. The resulting RNA pellet was washed twice with 70% ethanol (250 μl). RNA pellets were air-dried and then re-suspended in RNase-free water. Any potential genomic DNA contamination was removed by DNaseI treatment using the TURBO DNA-free Kit (Invitrogen).

## Reverse transcription-quantitative PCR (RT-qPCR)

RT-qPCR was performed using the CFX Opus 96 Real-Time PCR System (Bio-rad). To begin, total RNA samples were reverse transcribed into first-strand cDNA using the iScript Select cDNA Synthesis kit (Bio-rad). 1 μl of a 10X diluted cDNA sample was then used in the qPCR reaction with SsoAdvanced Universal SYBR Green Supermix (Bio-rad). Primers were designed using Primer3 software (v4.1.0). Relative mRNA expression levels were calculated using the $2^{-\triangle\triangle Ct}$ method, and the reference gene *act-1* was used to normalize the gene expression data.

## RNA-seq

Purified RNA was quantified and assessed for quality using an Epoch 2 microplate spectrophotometer (BioTek). cDNA libraries were prepared from 500 ng RNA per sample. Libraries were then sequenced on an Illumina HiSeq 2000 sequencing machine using paired-end sequencing with a read length of 100 nucleotides. Adaptor sequences and low-quality reads were removed using Trimmomatic, and the RNA-Seq reads were then mapped to the *C. elegans* genome using STAR 2.5.3a with default settings. Transcript abundance, measured as read counts per gene, was extracted using HTSeq. Differential gene expression analysis was performed using GFold[34]. Gene ontology (GO) analysis was conducted using DAVID.

## DAF-16 localization assay

The strain TJ356 (*daf-16::gfp*) was used to detect the intracellular DAF-16 localization. Synchronized L1 worms were transferred to peptone-free NGM plates covered with the lawns of plenty of *L. plantarum* only, *L. plantarum* with 0.2 μl of BW25113, *L. plantarum* with 0.2 μl of *E. coli pdxH* mutant, and *L. plantarum* with 0.2 μl of *E. coli pdxH* mutant as well as 1 mM PLP. The worms were placed at room temperature for 3 days, then collected and washed three times by M9 buffer. The worms were fixed on the 2% agarose pad of the slides, anesthetized with 0.25 mM levamisole, and covered with coverslips. The worms were imaged using a confocal scanning laser microscope. The percentages of DAF-16 localization (cytosolic, nuclear, and both) were calculated.

## Statistics and reproducibility

The statistical analyses were conducted using GraphPad Prism and Excel. Number of worms scored is indicated as *n*. The results are presented as the mean ± SEM, and the data were evaluated using an unpaired two-tailed Student's *t*-test. Data are represented as mean ± SEM. * Indicates *P*-value < 0.05, ** indicates *P*-value < 0.01, N.S. indicates non-significant difference. All data are representative of at least three independent experiments.

## Reporting summary

Further information on research design is available in the Nature Portfolio Reporting Summary linked to this article.

## Data availability

The data that support this finding are available from Supplementary Data 1; RNA-seq data is accessible on NCBI with GEO accession number GSE254343. All other data are available from the corresponding author on a reasonable request.

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

## Acknowledgements

This work was supported by the Texas A&M Startup grant from TEES and the Department of Chemical Engineering to Q.S. *C. elegans* strains were provided by the Caenorhabditis Genetics Center, which is funded by NIH Office of Research Infrastructure Programs (P40 OD010440).

## Author contributions

M.F. designed and performed the experiments. M.F., B.G. and D.R. collected and analyzed the data. M.F. and Q.S. wrote the paper. Q.S. and L.R.G. supervised the project and reviewed the manuscript.

## Competing interests

The authors declare no competing interests.
