## [Peer review file · Communications Biology]

Reviewers' comments:

Reviewer #1 (Remarks to the Author):

The paper by Min Feng et al shows that lack of vitamin B6 is the reason why *C. elegans* cannot develop on the probiotic bacteria *Lactiplantibacillus plantarum*.

The authors use a very nice screening strategy to identify the reason why a probiotic diet consisting of *Lactiplantibacillus plantarum* does not support development of *C. elegans* but rather causes larval arrest. Briefly, having identified that supplementation with the traditional *C. elegans* food source *E. coli* is sufficient to allow development on *Lactiplantibacillus plantarum*, they screened an *E. coli* deletion library consistent of nearly 4000 mutant strains, and isolated those that do not support normal development. Subsequent identification of the mutated genes and transcriptomic analysis leads to vitamin B6 being an essential component.

Probiotics are receiving increasing interest as dietary supplements and alternatives to traditional antibiotics and identifying the underlying molecular mechanisms is important and of general interest to the scientific community.

The paper is well written, easy to read and understand. The figures are clear and well presented. I really enjoyed reading the paper and I only have few reservations that should be addressed before publication.

1. The new *Lactobacillus* species names should be used and hence the strain ATCC8014 should be called *Lactiplantibacillus plantarum*.

2. It should be mentioned that *Lactiplantibacillus plantarum* has indeed been shown to have probiotic effects in *C. elegans* when fed to adults, see for example our work <https://doi.org/10.1038/s41598-021-89831-y>, [10.3389/fmicb.2022.886206](https://doi.org/10.1038/s41598-022-88620-6) and https://doi.org/10.1007/978-3-319-44703-2_18. Otherwise, the overall conclusion (L350-351) is not really supported.

3. The experiments appear to have been performed carefully. However, the number of replicate experiments and *n* in the individual experiments are not clear for all figures or the MM section. This information should be included.

4. Worm size / area needs to be quantified in figure 3C and 5E similar to the other figures. The described differences are not easy to see from the panels, since the worms in these are very small.

5. Whereas the screening part and verification of vitamin B6 of the paper is extremely convincing, the final model and conclusions regarding the mechanism causing arrest are too speculative. Additional experimental support needs to be provided.

i) L 231: Does DAF-16 indeed translocate to the nucleus when worms are fed *Lactiplantibacillus plantarum* and is this prevented by adding wt *E. coli* and not when adding *E. coli* mutants? This is a central point that can easily be addressed using the worm strain TJ356 and should be included.

ii) A general stress response could also cause nuclear translocation of DAF-16 – and such stress could

be caused by some of the other changed GOs identified. Thus, provided that DAF-16 does translocate to the nucleus, the effects of PLP and PN supplementation on DAF-16 localization should be tested to directly link these to DAF-16 and provide additional and more direct support of the model.

iii) If I understand the proposed model correctly, it predicts that *daf-16* mutants should not arrest on *Lactiplantibacillus plantarum*. This should be addressed using *daf-16* (*mu86*) null mutants. Including *daf-2*, *aak-2* as well as *tatn-1* mutants would also help strengthening the conclusions.

Minor

Line 1 is required?

L26 and L222 should be protein - DAF-16.

L144 nutrient signals, nutrients, signals, or all of them. I actually think the distinction is important and in this case all could be involved?

L222 *aak-2* should be in italic.

L227 *tatn-1* should be in italic.

L262 should the conclusion be part of the figure legend – I would delete it and only mention it in the main text.

L441 *act-1* should be in italic.

Figure 5A : the horizontal *pdxH* line is missing arrow heads.

L440 References: *C. elegans*, worm genes etc. should consistently be in italic.

L 436 Acknowledging the CGC will help maintain their funding. GCG asks to include the following statement: "Some strains were provided by the CGC, which is funded by NIH Office of Research Infrastructure Programs (P40 OD010440)."

Anders Olsen

Reviewer #2 (Remarks to the Author):

This manuscript by Feng et al., shows the effect on *C. elegans* growth of a diet of *L. plantarum*. The author convincingly show that traces amount of *E. coli* can supplement the nutritional deficit of this diet, and point to vitamin B6 as a key metabolite for postembryonic development. The approach, using an *E. coli* KO library is very useful and provide solid data on the pathway involved in the developmental defect. While the manuscript shows interesting data, the interpretation and claims of the authors are, at times, beyond the experimental evidence. For publication of this manuscript, my recommendation is to complete a couple of key experiments, or otherwise adapt the claims in the text.

Major concerns

- Relative to the nature of the developmental effect

Line 76: "...whereas worms fed solely on *L. plantarum* and developmentally arrested at early larval stage". The authors imply arrest of development, but this was not proven. It is difficult to differentiate arrest from developmental delay to the naked eye. However, the authors could use fluorescent reporters to analyze the state of the first divisions of postembryonic development, like seam cells and M cell. This way, it would be possible to assess if the observed effect corresponds to developmental arrest at a specific stage.

Line 22: "... the downstream metabolite pyridoxal 5-P (PLP, Vitamin B6) as essential nutritional factors initiating *C. elegans* postembryonic development"

Why talk about initiation of development?

Again, since it is not clear what is the stage the of the larvae on the *L. plantarum* diet (or the *pdx* mutants), it is difficult to confirm that Vit B6 is needed for initiating the process of postembryonic development.

In order to state that Vit B6 is necessary for initiation of development it is crucial to show that larvae are arrested before the first divisions of postembryonic development.

Figure 3B shows that larvae on the *L. plantarum* diet are larger than those on small amount of *E. coli*. In the same direction, Line 103: "the worms stayed at L1/L2 stage 5 days after been placed on the *L. plantarum* lawn". Here, the authors refer to L1/L2, is this different from the stage that is found the starved L1s are used to initiate the experiments?

Figure S2, bottom left picture: There are some larvae from the following generation, which are likely newly hatched L1. In the rest of the pictures, larvae are quite larger than those, indicating some growth and probably other developmental events took place.

- Relative to the effect of PLP.

PLP on its own does not restore growth, alive *E. coli* is also needed. This means the effect on development is not directly depending on PLP. Actually, the authors state this in line 185 "suggesting that the beneficial role of PLP for larval growth is likely dependent on some downstream bacterial usage of PLP."

However, in the model in Figure 6, and in further sections of the manuscript, the authors seem to imply a direct role of PLP. This should be clarified in the text.

This becomes especially relevant for the last part of the paper. In line 225, the authors say: "Previous studies have shown that TAT is a PLP-dependent enzyme that initiates the catabolism of tyrosine". From there, they hypothesized that the effect of PLP on development could be mediated by the activation of DAF-16 by low levels of TAT, via AAK-2 activation. However, given that the authors proved that supplementation with PLP is not sufficient for growth, this is not a strong hypothesis. While the *pdxH* mutant shows low levels of *tant-1*, the authors do not prove that a reduction in *tant-1* has the same effect on growth. Furthermore, they do not show any experiments to support a role of AAK-2 or DAF-16 in the effect of the *L. plantarum* diet.

Line 189: "The results show that the lack of *pdxH* does not affect worm growth (Figure 5F).

Do *pdxH* mutants completely lack Vit B6? Could *pdxH* mutants be making PLP from PL?

Do other *pdx* mutants also lead to growth when used in high amounts?

The RNA-seq results showed fluctuated gene transcription in host PLP binding.

- Relative to the role of DAF-16.

Line 25: "Additionally, bacterial PLP may act as a cofactor for host tyrosine aminotransferase, thereby promoting the translocation of *daf-16* to nucleus." This sentence in the abstract suggests that the authors have analysed DAF-16 localization, which is not the case.

Lines 219-231: This paragraph belongs to the Discussion, not the results section.

Furthermore, the interpretation by the author of the cited manuscript on *tant-1* seems to be wrong. The cited manuscript concludes, from the results in Figure 6E : "Together these findings suggest that the *tatn-1* enhancement of the eak dauer formation phenotype could be due to an increase in *daf-16*

transcriptional activity without an accompanying significant change in DAF-16 subcellular localization.

The model and figure legend 6E state: "A proposed model suggests that bacterial PLP acts as a cofactor of host TATN-1, promoting worm growth through DAF-16 translocation".

Does this mean translocation to the nucleus? This contradicts what is stated at the end of the results section and does not make sense in terms of what we know about DAF-16 activity.

According to the model, it seems that PLP leads to reduction of tant-1, when it should be the opposite. This is way the model seems to propose that PLP leads to nuclear localization of DAF-16, when it should be its absence what provokes that effect.

PLP is necessary to initiate development but then this sentence says that promotes translocation to the nucleus. Translocation of DAF-16 to the nucleus would arrest development, not promote it.

Minor and typos:

- Relative to the effect of *L. plantarum*

From <https://doi.org/10.3389/fnut.2022.1031502>

"Some species of microbiota lack the ability to biosynthesize vitamin B6, such as most genera within the Firmicutes phylum (*Veillonella*, *Ruminococcus*, *Faecalibacterium*, and *Lactobacillus* spp.)".

Line 131: "Thus, *E. coli* factors that rendered the *L. plantarum* edible are not secreted, heat nor UV stable." However, in the model in Figure 6, PLP seems to be secreted from the bacteria.

- Table 1 is not available for review.

- The number of independent replicates of the experiments is not stated in the text or legends.

- Others:

Line 26: *daf-16* should be DAF-16

Line 73: Revise the grammar of the sentence: "To investigated the worm growth on Gram-positive bacterium *L. plantarum*, a commonly consumed probiotic strain, and compare that with standard laboratory food *E. coli* OP50."

Line 76: Remove the "and at the end of the line.

Line 222: *aak-2* and *daf-16* should be AAK-2 and DAF-16

Line 222: "has a positive effect on DAF-16". Please state what is the effect.

Line 227: *tant-1* should be italicized

Line 365: "Then" should be "The"

Figure 3A. There is a typo on the Y axis legend mm3

Figure 4B. What is "WO"

Figure 5A. Arrowhead missing in the line between PLP and PMP.

Reply to Reviewers' comments:

Reviewer #1 (Remarks to the Author):

The paper by Min Feng et al shows that lack of vitamin B6 is the reason why *C. elegans* cannot develop on the probiotic bacteria *Lactiplantibacillus plantarum*.

The authors use a very nice screening strategy to identify the reason why a probiotic diet consisting of *Lactiplantibacillus plantarum* does not support development of *C. elegans* but rather causes larval arrest. Briefly, having identified that supplementation with the traditional *C. elegans* food source *E. coli* is sufficient to allow development on *Lactiplantibacillus plantarum*, they screened an *E. coli* deletion library consistent of nearly 4000 mutant strains, and isolated those that do not support normal development. Subsequent identification of the mutated genes and transcriptomic analysis leads to vitamin B6 being an essential component.

Probiotics are receiving increasing interest as dietary supplements and alternatives to traditional antibiotics and identifying the underlying molecular mechanisms is important and of general interest to the scientific community.

The paper is well written, easy to read and understand. The figures are clear and well presented. I really enjoyed reading the paper and I only have few reservations that should be addressed before publication.

1. The new Lactobacillus species names should be used and hence the strain ATCC8014 should be called *Lactiplantibacillus plantarum*.

Thank you for pointing this out. As of April 2020, the nomenclature for Lactobacillus species has been updated to Lactiplantibacillus. Thus, we changed "*Lactobacillus plantarum*" to "*Lactiplantibacillus plantarum*" throughout the manuscript.

2. It should be mentioned that *Lactiplantibacillus plantarum* has indeed been shown to have probiotic effects in *C. elegans* when fed to adults, see for example our work <https://doi.org/10.1038/s41598-021-89831-y>, [10.3389/fmicb.2022.886206](https://doi.org/10.3389/fmicb.2022.886206) and https://doi.org/10.1007/978-3-319-44703-2_18. Otherwise, the overall conclusion (L350-351) is not really supported.

Thank you for your suggestion. We incorporated the statement "*L. plantarum* has been shown to have probiotic effects in *C. elegans* when fed to adults *C. elegans*." into the introduction and have included those citations in line 60-61.

3. The experiments appear to have been performed carefully. However, the number of replicate experiments and n in the individual experiments are not clear for all figures or the MM section. This information should be included.

Thank you for pointing this out. We added the number of replicate experiments and n in the individual experiments in the figure legends. To be specific, "All data are representative of at least three independent experiments. n = number of worms scored. Data are represented as mean \pm

SEM. * indicates P-value < 0.05, ** indicates P-value < 0.01, N.S. indicates non-significant difference.” are included in 268-270, lines 280-281, lines 295-297, lines 320-322, and lines 334-336.

4. Worm size / area needs to be quantified in figure 3C and 5E similar to the other figures. The described differences are not easy to see from the panels, since the worms in these are very small. Thank you for pointing this out. We added the statistical analysis of the adult percentages to show the developmental difference in Figure 3C and 5E.

5. Whereas the screening part and verification of vitamin B6 of the paper is extremely convincing, the final model and conclusions regarding the mechanism causing arrest are too speculative. Additional experimental support needs to be provided.

i) L 231: Does DAF-16 indeed translocate to the nucleus when worms are fed *Lactiplantibacillus plantarum* and is this prevented by adding wt *E. coli* and not when adding *E. coli* mutants? This is a central point that can easily be addressed using the worm strain TJ356 and should be included. Thank you for your suggestion. We did new experiments with worm strain TJ356 to investigate DAF-16 translocation when worms are fed *L. plantarum* only, *L. plantarum* with wt *E. coli* (BW25113), and *L. plantarum* with *pdxH* mutant. The results, as shown in Figure 6E, indicated the increased DAF-16 nuclear translocation when fed *L. plantarum*. We have included the following paragraph in lines 241-251. “To shed light on the participation of DAF-16 in worm developmental arrest mediated by *L. plantarum*, we investigated the subcellular localization of DAF-16 in worms exposed to various conditions, including those exclusively fed *L. plantarum*, *L. plantarum* in combination with 0.2 µl of BW25113, *L. plantarum* with 0.2 µl of the *E. coli pdxH* mutant, and *L. plantarum* with 0.2 µl of the *E. coli pdxH* mutant in addition to 1 mM PLP. We found that worms fed *L. plantarum* only or *L. plantarum* with 0.2 µl of the *E. coli pdxH* mutant exhibited a significant increase in the nuclear localization of DAF-16 after 4 days incubation when compared to worms fed *L. plantarum* with 0.2 µl of wild-type *E. coli* BW25113 (Figure 6E). In addition, PLP supplementation reversed the DAF-16 nuclear translocation (Figure 6E), indicating the nuclear translocation of DAF-16 in worms fed *L. plantarum* with 0.2 µl of the *E. coli pdxH* mutant is likely due to stress associated with deficiency in PLP or a decrease in a downstream PLP-regulated process.”

ii) A general stress response could also cause nuclear translocation of DAF-16 – and such stress could be caused by some of the other changed GOs identified. Thus, provided that DAF-16 does translocate to the nucleus, the effects of PLP and PN supplementation on DAF-16 localization should be tested to directly link these to DAF-16 and provide additional and more direct support of the model.

Thank you for this suggestion. In order to rule out the possibility of the nuclear translocation due to a general stress from the environment, we supplemented PLP (the more downstream metabolites essential for worm growth) to worms fed *L. plantarum* with 0.2 µl of the *E. coli pdxH* mutant. The results, as shown in Figure 6E, indicated a similar nuclear translocation percentage as worms fed *L. plantarum* with 0.2 µl of the wild-type *E. coli* BW25113. We have added the following explanation in lines 248-251 “In addition, PLP supplementation reversed the DAF-16

nuclear translocation (Figure 6E), indicating the nuclear translocation of DAF-16 in worms fed *L. plantarum* with 0.2 μ l of the *E. coli pdxH* mutant is likely due to stress associated with deficiency in PLP or a decrease in a downstream PLP-regulated process.”

iii) If I understand the proposed model correctly, it predicts that *daf-16* mutants should not arrest on *Lactiplantibacillus plantarum*. This should be addressed using *daf-16* (*mu86*) null mutants. Including *daf-2*, *aak-2* as well as *tatn-1* mutants would also help strengthening the conclusions. Thank you for your suggestion. In the previous model, we proposed that DAF-16 nuclear translocated from cytosolic (DAF-16 activation) in the worm developmental arrest on *L. plantarum* in Figure 6E, in which case *daf-16* (*mu86*) null mutants should not arrest on *L. plantarum*. Thus, we performed experiments using *daf-16* (*mu86*) null mutants. However, when these mutants were exclusively fed *L. plantarum*, the worms arrested; and when fed *L. plantarum* with 0.2 μ l of the wild-type *E. coli* BW25113, they grew normally (Figure 6F). We added “To further clarify the role of *daf-16*, we asked whether *daf-16(mu86)* null mutants developmentally arrest or can develop on *L. plantarum*. When this mutant was exclusively fed *L. plantarum*, the worms exhibited developmental arrest (Figure 6F). This suggests that processes that lead to developmental arrest do not rely on *daf-16* for activation.” In lines 251-255.

This indicates that genes which induces developmental arrest do not need *daf-16* to activate them. DAF-16 does enter the nucleus upon *L. plantarum* feeding, but it’s not the reason for stalling developmental between L1 and L2. Developmental defect is possibly due to a lack of rate limiting factor for robust developmental progression. DAF-16 might just promote a coping mechanism to keep worms alive until they procure enough of that rate limiting substance. That rate limiting substance could either be vitamin B6 delivered via a bacteria-derived vehicle or some other *E. coli* factor(s) that require vitamin B6 for its/their synthesis. We have included “In this study, although DAF-16 does translocate into the nucleus upon *L. plantarum* feeding, it does not seem to be the primary cause of developmental arrest or stalling. The developmental defect from the *L. plantarum* diet is possibly due to a lack of rate-limiting factor crucial for robust developmental progression. DAF-16 might just promote a coping mechanism to sustain worm survival until an adequate amount of rate-limiting substance is acquired. This rate limiting substance could potentially be vitamin B6 delivered via a bacteria-derived vehicle or other *E. coli* factors that rely on vitamin B6 for their synthesis.” in the discussion in lines 410-417.

As the participation of *daf-2*, *aak-2*, and *tatn-1* are mostly predictive, we moved this hypothesis to the discussion in lines 368-380, “Interestingly, our RNA-seq analysis revealed multiple PLP binding genes are decreased in larval *C. elegans* fed with *L. plantarum* (Figure 6C), one of which is *tatn-1* (Figure 6C). *tatn-1* gene encodes a tyrosine aminotransferase (TAT), previous studies have shown that TAT is a PLP-dependent enzyme catalyzes conversion of tyrosine to 4-hydroxyphenylpyruvate, which is a rate-limiting step to metabolize tyrosine to fumarate and acetoacetate^{22,23}. RNAi knockdown of *tatn-1* gene expression can stall larval development^{24,25}. Here, we hypothesize that the PLP from *E. coli* functions as a coenzyme of worm TAT, the absence of PLP from the dietary bacteria leads to a reduced expression of *tatn-1* (Figure 6C), and subsequently leading to downstream deficiency in fumarate and acetoacetate or detrimental buildup of tyrosine and the developmental arrest. It is possible that *E. coli* is responsible for

delivering vitamin B6 to *C. elegans* TATN-1, either through a bacterial conjugate (such as a PLP-binding bacterial protein) or it is equally possible that *C. elegans* does not directly require vitamin B6, but instead it relies on specific rate-limiting molecules from *E. coli* that require vitamin B6 for their synthesis (Figure 6G).". We also revised the model based on the updated results (Figure 6G).

Minor

Line 1 is required?

Thank you for this suggestion. We added "is" in line 1 which is the title.

L26 and L222 should be protein - DAF-16.

Thank you for pointing this out. We corrected "daf-16" to "DAF-16" in line 26 and line 239.

L144 nutrient signals, nutrients, signals, or all of them. I actually think the distinction is important and in this case all could be involved?

Thank you for pointing this out. The distinction between nutrient signals, nutrients, signals is important. In our manuscript, we meant to discover the genetic components that is important to worm growth when fed *L. plantarum*. In the following section, we seek the nutrient signals and signaling pathways that could be involved. In line 158, they are "nutrients" in the context, thus we delete "signals".

L222 *aak-2* should be in italic.

Thank you for pointing this out. We corrected "aak-2" to "*aak-2*" in line 222.

L227 *tatn-1* should be in italic.

Thank you for pointing this out. We corrected "tatn-1" to "*tatn-1*" in line 369.

L262 should the conclusion be part of the figure legend – I would delete it and only mention it in the main text.

Thank you for pointing it out. We deleted this conclusion sentence "indicating that heat-stable, UV stable, and non-secreted factors from *E. coli* are required for normal worm growth" in line 262.

L441 *act-1* should be in italic.

Thank you for pointing this out. We corrected "act-1" to "*act-1*" in line 488.

Figure 5A : the horizontal pdxH line is missing arrow heads.

Thank you for pointing this out. We added the arrow head in Figure 5A.

L440 References: *C. elegans*, worm genes etc. should consistently be in italic.

Thank you for pointing this out. We have carefully checked and corrected the worm and gene names in the references.

L 436 Acknowledging the CGC will help maintain their funding. GCG asks to include the following statement: "Some strains were provided by the CGC, which is funded by NIH Office of Research Infrastructure Programs (P40 OD010440)."

Thank you for bringing this to our attention. We benefited a lot from CGC by getting all the strains from their center. We agree that acknowledging CGC is important. We have added "*C. elegans* strains were provided by the Caenorhabditis Genetics Center, which is funded by NIH Office of Research Infrastructure Programs (P40 OD010440)." to the acknowledgments in lines 512-513.

Anders Olsen

Reviewer #2 (Remarks to the Author):

This manuscript by Feng et al., shows the effect on *C. elegans* growth of a diet of *L. plantarum*. The author convincingly show that traces amount of *E. coli* can supplement the nutritional deficit of this diet, and point to vitamin B6 as a key metabolite for postembryonic development. The approach, using an *E. coli* KO library is very useful and provide solid data on the pathway involved in the developmental defect. While the manuscript shows interesting data, the interpretation and claims of the authors are, at times, beyond the experimental evidence. For publication of this manuscript, my recommendation is to complete a couple of key experiments, or otherwise adapt the claims in the text.

Major concerns

- Relative to the nature of the developmental effect

Line 76: "...whereas worms fed solely on *L. plantarum* and developmentally arrested at early larval stage". The authors imply arrest of development, but this was not proven. It is difficult to differentiate arrest from developmental delay to the naked eye. However, the authors could use fluorescent reporters to analyze the state of the first divisions of postembryonic development, like seam cells and M cell. This way, it would be possible to assess if the observed effect corresponds to developmental arrest at a specific stage.

Thank you for your suggestion. We conducted new experiments using M lineage patterning as an indicator of worm development progression using *hlh-8::gfp* reporter expressed in the M cell. The result indicated that the development of worms fed with *L. plantarum* arrested at late L1 to L2 stage. We have added the results in Figure 1E and added the following description in lines 91-102 "To further elucidate the stage at which developmental arrest occurred in the presence of *L. plantarum*, we assessed M lineage patterning as an indicator of worm development progression. In postembryonic development, a single mesodermal blast cell (M) undergoes division to produce a small number of additional mesodermal cells. In hermaphrodites, the M divisions occurring in early larval development result in the formation of 14 striated body wall muscles, two sex myoblasts (SMs), and two coelomocytes. By the L4 stage, the SMs undergo division, resulting in 16 SM descendants located near the vulval opening. Here, we used *hlh-8::gfp* reporter expressed in the M-cell as the indicator. We observed that the GFP signal displayed the pattern characteristic

of the stage between the late L1 stage to L2 stage in worms fed *L. plantarum* after 3 days, while worms fed OP50 exhibited a GFP signal pattern resembling the L4 stage (Figure 1E). This observation strongly suggests that the development of worms fed with *L. plantarum* arrested or stalled at the late L1 stage to L2 stage.”

Line 22: “... the downstream metabolite pyridoxal 5-P (PLP, Vitamin B6) as essential nutritional factors initiating *C. elegans* postembryonic development” Why talk about initiation of development? Again, since it is not clear what is the stage the of the larvae on the *L. plantarum* diet (or the *pdx* mutants), it is difficult to confirm that Vit B6 is needed for initiating the process of postembryonic development. In order to state that Vit B6 is necessary for initiation of development it is crucial to show that larvae are arrested before the first divisions of postembryonic development.

Thank you for pointing this out. We talked about initiation of development because we thought the worms could be arrested at L1 stage when we wrote the draft. To avoid misleading the reader, we using M lineage patterning as an indicator of worm developmental stage. The new result suggested the worms arrested at late L1 stage to L2 stage. We have changed “initiation of development” to “development” throughout the manuscript.

Figure 3B shows that larvae on the *L. plantarum* diet are larger than those on small amount of *E. coli*. In the same direction, Line 103: “the worms stayed at L1/L2 stage 5 days after been placed on the *L. plantarum* lawn”. Here, the authors refer to L1/L2, is this different from the stage that is found the starved L1s are used to initiate the experiments? Figure S2, bottom left picture: There are some larvae from the following generation, which are likely newly hatched L1. In the rest of the pictures, larvae are quite larger than those, indicating some growth and probably other developmental events took place.

Thank you for pointing this out. Yes, the worms fed *L. plantarum* are slightly bigger than starved L1s. Per your suggestion, we used M cell division to assess the stage of those arrested worms. The result indicated the development of worms fed with *L. plantarum* arresting at the L2 stage. We have added the results in Figure 1E and added the following paragraph in lines 91-102 “To further elucidate the stage at which developmental arrest occurred in the presence of *L. plantarum*, we assessed M lineage patterning as an indicator of worm development progression. In postembryonic development, a single mesodermal blast cell (M) undergoes division to produce a small number of additional mesodermal cells. In hermaphrodites, the M divisions occurring in early larval development result in the formation of 14 striated body wall muscles, two sex myoblasts (SMs), and two coelomocytes. By the L4 stage, the SMs undergo division, resulting in 16 SM descendants located near the vulval opening. Here, we used *hlh-8::gfp* reporter expressed in the M-cell as the indicator. We observed that the GFP signal displayed the pattern characteristic of the stage between the late L1 stage to L2 stage in worms fed *L. plantarum* after 3 days, while worms fed OP50 exhibited a GFP signal pattern resembling the L4 stage (Figure 1E). This observation strongly suggests that the development of worms fed with *L. plantarum* arrested or stalled at the late L1 stage to L2 stage.”

- Relative to the effect of PLP.

PLP on its own does not restore growth, alive *E. coli* is also needed. This means the effect on development is not directly depending on PLP. Actually, the authors state this in line 185 “suggesting that the beneficial role of PLP for larval growth is likely dependent on some downstream bacterial usage of PLP.” However, in the model in Figure 6, and in further sections of the manuscript, the authors seem to imply a direct role of PLP. This should be clarified in the text. Thank you for your suggestion. We agree with the reviewer. From Figure 5E, we concluded PLP on its own does not restore growth - alive *E. coli pdxH* mutant is also needed. To make it clearer, we changed the sentence to “these results demonstrate that *E. coli pdxH* mutant provides limited nutrition, with vitamin B6 deriving from its prior culturing conditions, supporting worm growth.” in lines 211-213. In this section, we also included “We found that supplementing 1mM PLP without live *pdxH* mutant did not promote the worm growth (Figure 5E), suggesting that the beneficial role of PLP for larval growth either requires some live *E. coli* molecules to deliver the PLP to *C. elegans*, or *C. elegans* is dependent on some downstream bacterial usage of PLP.” in lines 197-201, “The *pdxH* mutant was grown in rich nutrient containing yeast extract, which provided the vitamins required for the mutant’s enzyme functions.” in lines 203-204.

This becomes especially relevant for the last part of the paper. In line 225, the authors say: “Previous studies have shown that TAT is a PLP-dependent enzyme that initiates the catabolism of tyrosine”. From there, they hypothesized that the effect of PLP on development could be mediated by the activation of DAF-16 by low levels of TAT, via AAK-2 activation. However, given that the authors proved that supplementation with PLP is not sufficient for growth, this is not a strong hypothesis. While the *pdxH* mutant shows low levels of tant-1, the authors do not prove that a reduction in tant-1 has the same effect on growth. Furthermore, they do not show any experiments to support a role of AAK-2 or DAF-16 in the effect of the *L. plantarum* diet.

Thank you for pointing this out. In order to demonstrate the role of *daf-16* in the effect of the *L. plantarum* diet and raise our hypothesis, we did more experiments by using worm strain TJ356 (*daf-16::gfp*) to investigate DAF-16 translocation when worms are fed *L. plantarum* only, *L. plantarum* with wt *E. coli* (BW25113), and *L. plantarum* with *pdxH* mutant. The results are showing in Figure 6E, indicating the increased DAF-16 nuclear translocation when fed *L. plantarum*, including “To shed light on the participation of DAF-16 in worm developmental arrest mediated by *L. plantarum*, we investigated the subcellular localization of DAF-16 in worms exposed to various conditions, including those exclusively fed *L. plantarum*, *L. plantarum* in combination with 0.2 μ l of BW25113, *L. plantarum* with 0.2 μ l of the *E. coli pdxH* mutant, and *L. plantarum* with 0.2 μ l of the *E. coli pdxH* mutant in addition to 1 mM PLP. We found that worms fed *L. plantarum* only or *L. plantarum* with 0.2 μ l of the *E. coli pdxH* mutant exhibited a significant increase in the nuclear localization of DAF-16 after 4 days incubation when compared to worms fed *L. plantarum* with 0.2 μ l of wild-type *E. coli* BW25113 (Figure 6E). In addition, PLP supplementation reversed the DAF-16 nuclear translocation (Figure 6E), indicating the nuclear translocation of DAF-16 in worms fed *L. plantarum* with 0.2 μ l of the *E. coli pdxH* mutant is likely due to stress associated with deficiency in PLP or a decrease in a downstream PLP-regulated process.” in line 241-251.

In addition, we investigated the growth of *daf-16 (mu86)* null worm mutant on the *L. plantarum* diet. Result is included in Figure 6F, “To further clarify the role of *daf-16*, we asked whether *daf-*

16(mu86) null mutants developmentally arrest or can develop on *L. plantarum*. When this mutant was exclusively fed *L. plantarum*, the worms exhibited developmental arrest (Figure 6F). This suggests that processes that lead to developmental arrest do not rely on *daf-16* for activation.” In line 251-255. Therefore, we revised the discussion based on those new results, including “In this study, although DAF-16 does translocate into the nucleus upon *L. plantarum* feeding, it does not seem to be the primary cause of developmental arrest or stalling. The developmental defect from the *L. plantarum* diet is possibly due to a lack of rate-limiting factor crucial for robust developmental progression. DAF-16 might just promote a coping mechanism to sustain worm survival until an adequate amount of rate-limiting substance is acquired. This rate limiting substance could potentially be vitamin B6 delivered via a bacteria-derived vehicle or other *E. coli* factors that rely on vitamin B6 for their synthesis.” In lines 410-417.

As for the hypothesis on *tatn-1*, we added further references showing knockdown of *tatn-1* gene expression can stall larval development {24,25} and moved this part to the discussion, including “Interestingly, our RNA-seq analysis revealed multiple PLP binding genes are decreased in larval *C. elegans* fed with *L. plantarum* (Figure 6C), one of which is *tatn-1* (Figure 6C). *tatn-1* gene encodes a tyrosine aminotransferase (TAT), previous studies have shown that TAT is a PLP-dependent enzyme catalyzes conversion of tyrosine to 4-hydroxyphenylpyruvate, which is a rate-limiting step to metabolize tyrosine to fumarate and acetoacetate^{22,23}. RNAi knockdown of *tatn-1* gene expression can stall larval development^{24,25}. Here, we hypothesize that the PLP from *E. coli* functions as a coenzyme of worm TAT, the absence of PLP from the dietary bacteria leads to a reduced expression of *tatn-1* (Figure 6C), and subsequently leading to downstream deficiency in fumarate and acetoacetate or detrimental buildup of tyrosine and the developmental arrest. It is possible that *E. coli* is responsible for delivering vitamin B6 to *C. elegans* TATN-1, either through a bacterial conjugate (such as a PLP-binding bacterial protein) or it is equally possible that *C. elegans* does not directly require vitamin B6, but instead it relies on specific rate-limiting molecules from *E. coli* that require vitamin B6 for their synthesis (Figure 6G).” in lines 368-380.

Line 189: “The results show that the lack of *pdxH* does not affect worm growth (Figure 5F).

Do *pdxH* mutants completely lack Vit B6? Could *pdxH* mutants be making PLP from PL?

Do other *pdx* mutants also lead to growth when used in high amounts?

Thank you for bringing this to our attention. Lack of *pdxH* does not affect worm growth with *E. coli* as the diet alone, which means that the function of *pdxH* can be compensated for by increasing the amount of the *E. coli* mutant. It could be some salvage pathway that makes PLP from PL. To stress the role of PLP from *E. coli* that promotes *L. plantarum* usage, we included “The *pdxH* mutant was grown in rich nutrient containing yeast extract, which provided the vitamins required for the mutant’s enzyme functions. Taken together, these results demonstrate that *E. coli pdxH* mutant provides limited nutrition, with vitamin B6 deriving from its prior culturing conditions, supporting worm growth.” in lines 210-213. “Lack of autonomous *E. coli* synthesis of PLP does not affect worm growth as the diet alone, provided sufficient live mutant bacteria is fed to the worms (Figure 5F). This observation suggests that PLP-binding enzymes in living *pdxH* mutants can either (i) serve as a delivery vehicle for the vitamin to *C. elegans* enzymes, such as tyrosine amino transferase, or (ii) vitamin B6 might be required for *E. coli* PLP-dependent enzymes to synthesize essential molecules, such as amino acids, that might be deficient in the context of a

L. plantarum diet for larval *C. elegans*³⁰. When *pdxH* mutant is cultivated in nutritionally rich LB broth (usually containing yeast extract), which likely provides vitamin B6 along with all of the amino acids that are synthesized by vitamin B6 binding enzymes. It's possible that the essential enzymes in *pdxH* mutant diet are loaded with vitamin B6 from its yeast extract containing growth media. Consequently, the greater the quantity of *pdxH* mutant added to *C. elegans*, the more indirectly yeast-derived vitamin B6 is introduced, thereby supporting worm growth." in lines 388-400 in the discussion.

The RNA-seq results showed fluctuated gene transcription in host PLP binding.

- Relative to the role of DAF-16.

Line 25: "Additionally, bacterial PLP may act as a cofactor for host tyrosine aminotransferase, thereby promoting the translocation of *daf-16* to nucleus." This sentence in the abstract suggests that the authors have analysed DAF-16 localization, which is not the case.

Thank you for pointing this out. We revised the abstract based on new results, includes "Additionally, the developmental arrest induced by the *L. plantarum* diet in worm does not depend on the activation of FoxO/DAF-16." in lines 25-26.

Lines 219-231: This paragraph belongs to the Discussion, not the results section.

Thank you for this suggestion. We moved this paragraph to the discussion.

Furthermore, the interpretation by the author of the cited manuscript on *tatn-1* seems to be wrong. The cited manuscript concludes, from the results in Figure 6E: "Together these findings suggest that the *tatn-1* enhancement of the eak dauer formation phenotype could be due to an increase in *daf-16* transcriptional activity without an accompanying significant change in DAF-16 subcellular localization. The model and figure legend 6E state: "A proposed model suggests that bacterial PLP acts as a cofactor of host TATN-1, promoting worm growth through DAF-16 translocation". Does this mean translocation to the nucleus? This contradicts what is stated at the end of the results section and does not make sense in terms of what we know about DAF-16 activity. According to the model, it seems that PLP leads to reduction of *tatn-1*, when it should be the opposite. This is way the model seems to propose that PLP leads to nuclear localization of DAF-16, when it should be its absence what provokes that effect. PLP is necessary to initiate development but then this sentence says that promotes translocation to the nucleus. Translocation of DAF-16 to the nucleus would arrest development, not promote it.

Thank you for pointing this out. In order to demonstrate the role of *daf-16* in the effect of the *L. plantarum* diet and raise our hypothesis, we did more experiments by using worm strain TJ356 (*daf-16::gfp*) to investigate DAF-16 translocation when worms are fed *L. plantarum* only, *L. plantarum* with wt *E. coli* (BW25113), and *L. plantarum* with *pdxH* mutant. The results are showing in Figure 6E, indicating the increased DAF-16 nuclear translocation when fed *L. plantarum*, including "To shed light on the participation of DAF-16 in worm developmental arrest mediated by *L. plantarum*, we investigated the subcellular localization of DAF-16 in worms exposed to various conditions, including those exclusively fed *L. plantarum*, *L. plantarum* in combination with 0.2 μ l of BW25113, *L. plantarum* with 0.2 μ l of the *E. coli pdxH* mutant, and *L. plantarum* with 0.2 μ l of the *E. coli pdxH* mutant in addition to 1 mM PLP. We found that worms

fed *L. plantarum* only or *L. plantarum* with 0.2 µl of the *E. coli pdxH* mutant exhibited a significant increase in the nuclear localization of DAF-16 after 4 days incubation when compared to worms fed *L. plantarum* with 0.2 µl of wild-type *E. coli* BW25113 (Figure 6E). In addition, PLP supplementation reversed the DAF-16 nuclear translocation (Figure 6E), indicating the nuclear translocation of DAF-16 in worms fed *L. plantarum* with 0.2 µl of the *E. coli pdxH* mutant is likely due to stress associated with deficiency in PLP or a decrease in a downstream PLP-regulated process.” in line 241-251.

In addition, we investigated the growth of *daf-16 (mu86)* null worm mutant on the *L. plantarum* diet. Result is included in Figure 6F, “To further clarify the role of *daf-16*, we asked whether *daf-16(mu86)* null mutants developmentally arrest or can develop on *L. plantarum*. When this mutant was exclusively fed *L. plantarum*, the worms exhibited developmental arrest (Figure 6F). This suggests that processes that lead to developmental arrest do not rely on *daf-16* for activation.” In line 251-255. Therefore, we revised the discussion based on those new results, including “In this study, although DAF-16 does translocate into the nucleus upon *L. plantarum* feeding, it does not seem to be the primary cause of developmental arrest or stalling. The developmental defect from the *L. plantarum* diet is possibly due to a lack of rate-limiting factor crucial for robust developmental progression. DAF-16 might just promote a coping mechanism to sustain worm survival until an adequate amount of rate-limiting substance is acquired. This rate limiting substance could potentially be vitamin B6 delivered via a bacteria-derived vehicle or other *E. coli* factors that rely on vitamin B6 for their synthesis.” In lines 410-417.

As for the hypothesis on *tatn-1*, we added further references showing knockdown of *tatn-1* gene expression can stall larval development {24,25} and moved this part to the discussion, including “Interestingly, our RNA-seq analysis revealed multiple PLP binding genes are decreased in larval *C. elegans* fed with *L. plantarum* (Figure 6C), one of which is *tatn-1* (Figure 6C). *tatn-1* gene encodes a tyrosine aminotransferase (TAT), previous studies have shown that TAT is a PLP-dependent enzyme catalyzes conversion of tyrosine to 4-hydroxyphenylpyruvate, which is a rate-limiting step to metabolize tyrosine to fumarate and acetoacetate^{22,23}. RNAi knockdown of *tatn-1* gene expression can stall larval development^{24,25}. Here, we hypothesize that the PLP from *E. coli* functions as a coenzyme of worm TAT, the absence of PLP from the dietary bacteria leads to a reduced expression of *tatn-1* (Figure 6C), and subsequently leading to downstream deficiency in fumarate and acetoacetate or detrimental buildup of tyrosine and the developmental arrest. It is possible that *E. coli* is responsible for delivering vitamin B6 to *C. elegans* TATN-1, either through a bacterial conjugate (such as a PLP-binding bacterial protein) or it is equally possible that *C. elegans* does not directly require vitamin B6, but instead it relies on specific rate-limiting molecules from *E. coli* that require vitamin B6 for their synthesis (Figure 6G).” in lines 368-380.

Minor and typos:

- Relative to the effect of *L. plantarum*

From <https://doi.org/10.3389/fnut.2022.1031502>

“Some species of microbiota lack the ability to biosynthesize vitamin B6, such as most genera within the Firmicutes phylum (Veillonella, Ruminococcus, Faecalibacterium, and Lactobacillus spp.)”.

Thank you for this suggestion. We add this to the discussion “Some species of microbiota including Lactobacillus lack the ability to biosynthesize vitamin B6, which makes synergistic in providing nutrients to the host important between microbes.” in lines 383-385.

Line 131: “Thus, E. coli factors that rendered the L. plantarum edible are not secreted, heat nor UV stable.” However, in the model in Figure 6, PLP seems to be secreted from the bacteria.

Thank you for pointing this out. As indicated in the rightmost image in Figure 3C, worms can not grow with E. coli conditioned media, thus we concluded the factor that rendered the L. plantarum edible are not secreted. To avoid confusion, we modified the arrow origin in Figure 6G from outside to inside the bacteria.

- Table 1 is not available for review.

Thank you for pointing this out. We uploaded the Table 1.

- The number of independent replicates of the experiments is not stated in the text or legends.

Thank you for pointing this out. We added the number of replicate experiments and n in the individual experiments in the figure legends. To be specific, “All data are representative of at least three independent experiments. n = number of worms scored. Data are represented as mean \pm SEM. * indicates P-value < 0.05, ** indicates P-value < 0.01, N.S. indicates non-significant difference.” are included in lines 268-270, lines 280-281, lines 295-297, lines 320-322, and lines 334-336.

- Others:

Line 26: daf-16 should be DAF-16

Thank you for pointing this out. We changed “daf-16” to “DAF-16” in line 26.

Line 73: Revise the grammar of the sentence: “To investigated the worm growth on Gram-positive bacterium L. plantarum, a commonly consumed probiotic strain, and compare that with standard laboratory food E. coli OP50.”

Thank you for pointing this out. We revised this sentence to “To investigate the effect of two different food sources, a commonly consumed probiotic Gram-positive bacterium L. Plantarum and the standard laboratory food E. coli OP50, we fed synchronized L1 worms with L. plantarum and E. coli OP50 individually and measure worm development.” in line 73-75.

Line 76: Remove the “and at the end of the line.

Thank you for the suggestion. We removed the “and” in line 76.

Line 222: aak-2 and daf-16 should be AAK-2 and DAF-16

Thank you for pointing this out. We changed “aak-2” and “daf-16” to “AAK-2” and “DAF-16” in line 239.

Line 222: “has a positive effect on DAF-16”. Please state what is the effect.
Thank you for pointing this out. We revised this to “activates DAF-16”.

Line 227: *tatn-1* should be italicized
Thank you for pointing this out. “*tatn-1*” is now italic in line 369.

Line 365: “Then” should be “The”
Thank you for pointing this out. We changed “Then” to “The” in line 365.

Figure 3A. There is a typo on the Y axis legend mm³
Thank you for pointing this out. We changed “mm³” to “μm³” in Figure 3A

Figure 4B. What is “WO”
Thank you for pointing this out. This should be a no bacteria feeding control. We changed it to “NO” in Figure 4B.

Figure 5A. Arrowhead missing in the line between PLP and PMP.
Thank you for pointing this out. The *pdxH* is necessary for PMP conversion to PLP. We added the arrow head in Figure 5A.

REVIEWERS' COMMENTS:

Reviewer #1 (Remarks to the Author):

In my opinion, the authors have satisfactorily addressed the concerns raised by the reviewers in the revised manuscript. In particular, the additional experiments regarding the role of DAF-16 have made this part of the paper less speculative. I think this improved manuscript is ready for publication.

Reviewer #2 (Remarks to the Author):

The authors have replied to the concerns raised during the first round of review. Nevertheless, there are a couple of minor changes to implement:

Line 20: Revise the sentence "that are indispensable for *C. elegans* larval growth on original(ly?) not nutritionally sufficient bacteria *L. plantarum*"

Line 67: Revise the sentence "thereby contribute(ing) to the host development."

Lines 147-9: The sentence in the text is "Here we sought to determine whether a diet supplemented with *L. plantarum* could modify the epigenetic patterns of *C. elegans* and render the diet edible for their offspring."

Since there is nothing tested relative to the epigenetic patterns, the sentence should be "Here we sought to determine whether a diet supplemented with *L. plantarum* could render the diet edible for their offspring."

Reply to Reviewers' comments:

Second round

Reviewer #1 (Remarks to the Author):

In my opinion, the authors have satisfactorily addressed the concerns raised by the reviewers in the revised manuscript. In particular, the additional experiments regarding the role of DAF-16 have made this part of the paper less speculative. I think this improved manuscript is ready for publication.

Thank you for your insightful comments and valuable improvements to our manuscript.

Reviewer #2 (Remarks to the Author):

The authors have replied to the concerns raised during the first round of review.

Thank you for your insightful comments and valuable improvements to our manuscript.

Nevertheless, there are a couple of minor changes to implement:

Line 20: Revise the sentence “that are indispensable for *C. elegans* larval growth on original(ly?) not nutritionally sufficient bacteria *L. plantarum*”

Thank you for pointing this out. We corrected “original” to “originally” in line 20.

Line 67: Revise the sentence “thereby contribute(ing) to the host development.

Thank you for pointing this out. We corrected “contribute” to “contributing” in line 68.

Lines 147-9: The sentence in the text is “Here we sought to determine whether a diet supplemented with *L. plantarum* could modify the epigenetic patterns of *C. elegans* and render the diet edible for their offspring.” Since there is nothing tested relative to the epigenetic patterns, the sentence should be “Here we sought to determine whether a diet supplemented with *L. plantarum* could render the diet edible for their offspring.”

Thank you for your suggestion. We deleted this sentence “modify the epigenetic patterns of *C. elegans* and.” In line 148.

First round

Reviewer #1 (Remarks to the Author):

The paper by Min Feng et al shows that lack of vitamin B6 is the reason why *C. elegans* cannot develop on the probiotic bacteria *Lactiplantibacillus plantarum*.

The authors use a very nice screening strategy to identify the reason why a probiotic diet consisting of *Lactiplantibacillus plantarum* does not support development of *C. elegans* but rather

causes larval arrest. Briefly, having identified that supplementation with the traditional *C. elegans* food source *E. coli* is sufficient to allow development on *Lactiplantibacillus plantarum*, they screened an *E. coli* deletion library consistent of nearly 4000 mutant strains, and isolated those that do not support normal development. Subsequent identification of the mutated genes and transcriptomic analysis leads to vitamin B6 being an essential component.

Probiotics are receiving increasing interest as dietary supplements and alternatives to traditional antibiotics and identifying the underlying molecular mechanisms is important and of general interest to the scientific community.

The paper is well written, easy to read and understand. The figures are clear and well presented. I really enjoyed reading the paper and I only have few reservations that should be addressed before publication.

1. The new *Lactobacillus* species names should be used and hence the strain ATCC8014 should be called *Lactiplantibacillus plantarum*.

Thank you for pointing this out. As of April 2020, the nomenclature for *Lactobacillus* species has been updated to *Lactiplantibacillus*. Thus, we changed "*Lactobacillus plantarum*" to "*Lactiplantibacillus plantarum*" throughout the manuscript.

2. It should be mentioned that *Lactiplantibacillus plantarum* has indeed been shown to have probiotic effects in *C. elegans* when fed to adults, see for example our work <https://doi.org/10.1038/s41598-021-89831-y>, [10.3389/fmicb.2022.886206](https://doi.org/10.3389/fmicb.2022.886206) and https://doi.org/10.1007/978-3-319-44703-2_18. Otherwise, the overall conclusion (L350-351) is not really supported.

Thank you for your suggestion. We incorporated the statement "*L. plantarum* has been shown to have probiotic effects in *C. elegans* when fed to adults *C. elegans*." into the introduction and have included those citations in line 60-61.

3. The experiments appear to have been performed carefully. However, the number of replicate experiments and n in the individual experiments are not clear for all figures or the MM section. This information should be included.

Thank you for pointing this out. We added the number of replicate experiments and n in the individual experiments in the figure legends. To be specific, "All data are representative of at least three independent experiments. n = number of worms scored. Data are represented as mean \pm SEM. * indicates P-value < 0.05, ** indicates P-value < 0.01, N.S. indicates non-significant difference." are included in 268-270, lines 280-281, lines 295-297, lines 320-322, and lines 334-336.

4. Worm size / area needs to be quantified in figure 3C and 5E similar to the other figures. The described differences are not easy to see from the panels, since the worms in these are very small. Thank you for pointing this out. We added the statistical analysis of the adult percentages to show the developmental difference in Figure 3C and 5E.

5. Whereas the screening part and verification of vitamin B6 of the paper is extremely convincing, the final model and conclusions regarding the mechanism causing arrest are too speculative. Additional experimental support needs to be provided.

i) L 231: Does DAF-16 indeed translocate to the nucleus when worms are fed *Lactiplantibacillus plantarum* and is this prevented by adding wt *E. coli* and not when adding *E. coli* mutants? This is a central point that can easily be addressed using the worm strain TJ356 and should be included. Thank you for your suggestion. We did new experiments with worm strain TJ356 to investigate DAF-16 translocation when worms are fed *L. plantarum* only, *L. plantarum* with wt *E. coli* (BW25113), and *L. plantarum* with *pdxH* mutant. The results, as shown in Figure 6E, indicated the increased DAF-16 nuclear translocation when fed *L. plantarum*. We have included the following paragraph in lines 241-251. “To shed light on the participation of DAF-16 in worm developmental arrest mediated by *L. plantarum*, we investigated the subcellular localization of DAF-16 in worms exposed to various conditions, including those exclusively fed *L. plantarum*, *L. plantarum* in combination with 0.2 µl of BW25113, *L. plantarum* with 0.2 µl of the *E. coli pdxH* mutant, and *L. plantarum* with 0.2 µl of the *E. coli pdxH* mutant in addition to 1 mM PLP. We found that worms fed *L. plantarum* only or *L. plantarum* with 0.2 µl of the *E. coli pdxH* mutant exhibited a significant increase in the nuclear localization of DAF-16 after 4 days incubation when compared to worms fed *L. plantarum* with 0.2 µl of wild-type *E. coli* BW25113 (Figure 6E). In addition, PLP supplementation reversed the DAF-16 nuclear translocation (Figure 6E), indicating the nuclear translocation of DAF-16 in worms fed *L. plantarum* with 0.2 µl of the *E. coli pdxH* mutant is likely due to stress associated with deficiency in PLP or a decrease in a downstream PLP-regulated process.”

ii) A general stress response could also cause nuclear translocation of DAF-16 – and such stress could be caused by some of the other changed GOs identified. Thus, provided that DAF-16 does translocate to the nucleus, the effects of PLP and PN supplementation on DAF-16 localization should be tested to directly link these to DAF-16 and provide additional and more direct support of the model.

Thank you for this suggestion. In order to rule out the possibility of the nuclear translocation due to a general stress from the environment, we supplemented PLP (the more downstream metabolites essential for worm growth) to worms fed *L. plantarum* with 0.2 µl of the *E. coli pdxH* mutant. The results, as shown in Figure 6E, indicated a similar nuclear translocation percentage as worms fed *L. plantarum* with 0.2 µl of the wild-type *E. coli* BW25113. We have added the following explanation in lines 248-251 “In addition, PLP supplementation reversed the DAF-16 nuclear translocation (Figure 6E), indicating the nuclear translocation of DAF-16 in worms fed *L. plantarum* with 0.2 µl of the *E. coli pdxH* mutant is likely due to stress associated with deficiency in PLP or a decrease in a downstream PLP-regulated process.”

iii) If I understand the proposed model correctly, it predicts that *daf-16* mutants should not arrest on *Lactiplantibacillus plantarum*. This should be addressed using *daf-16* (*mu86*) null mutants. Including *daf-2*, *aak-2* as well as *tatn-1* mutants would also help strengthening the conclusions.

Thank you for your suggestion. In the previous model, we proposed that DAF-16 nuclear translocated from cytosolic (DAF-16 activation) in the worm developmental arrest on *L.*

plantarum in Figure 6E, in which case *daf-16* (*mu86*) null mutants should not arrest on *L. plantarum*. Thus, we performed experiments using *daf-16* (*mu86*) null mutants. However, when these mutants were exclusively fed *L. plantarum*, the worms arrested; and when fed *L. plantarum* with 0.2 μ l of the wild-type *E. coli* BW25113, they grew normally (Figure 6F). We added “To further clarify the role of *daf-16*, we asked whether *daf-16(mu86)* null mutants developmentally arrest or can develop on *L. plantarum*. When this mutant was exclusively fed *L. plantarum*, the worms exhibited developmental arrest (Figure 6F). This suggests that processes that lead to developmental arrest do not rely on *daf-16* for activation.” In lines 251-255.

This indicates that genes which induces developmental arrest do not need *daf-16* to activate them. DAF-16 does enter the nucleus upon *L. plantarum* feeding, but it’s not the reason for stalling developmental between L1 and L2. Developmental defect is possibly due to a lack of rate limiting factor for robust developmental progression. DAF-16 might just promote a coping mechanism to keep worms alive until they procure enough of that rate limiting substance. That rate limiting substance could either be vitamin B6 delivered via a bacteria-derived vehicle or some other *E. coli* factor(s) that require vitamin B6 for its/their synthesis. We have included “In this study, although DAF-16 does translocate into the nucleus upon *L. plantarum* feeding, it does not seem to be the primary cause of developmental arrest or stalling. The developmental defect from the *L. plantarum* diet is possibly due to a lack of rate-limiting factor crucial for robust developmental progression. DAF-16 might just promote a coping mechanism to sustain worm survival until an adequate amount of rate-limiting substance is acquired. This rate limiting substance could potentially be vitamin B6 delivered via a bacteria-derived vehicle or other *E. coli* factors that rely on vitamin B6 for their synthesis.” in the discussion in lines 410-417.

As the participation of *daf-2*, *aak-2*, and *tatn-1* are mostly predictive, we moved this hypothesis to the discussion in lines 368-380, “Interestingly, our RNA-seq analysis revealed multiple PLP binding genes are decreased in larval *C. elegans* fed with *L. plantarum* (Figure 6C), one of which is *tatn-1* (Figure 6C). *tatn-1* gene encodes a tyrosine aminotransferase (TAT), previous studies have shown that TAT is a PLP-dependent enzyme catalyzes conversion of tyrosine to 4-hydroxyphenylpyruvate, which is a rate-limiting step to metabolize tyrosine to fumarate and acetoacetate^{22,23}. RNAi knockdown of *tatn-1* gene expression can stall larval development^{24,25}. Here, we hypothesize that the PLP from *E. coli* functions as a coenzyme of worm TAT, the absence of PLP from the dietary bacteria leads to a reduced expression of *tatn-1* (Figure 6C), and subsequently leading to downstream deficiency in fumarate and acetoacetate or detrimental buildup of tyrosine and the developmental arrest. It is possible that *E. coli* is responsible for delivering vitamin B6 to *C. elegans* TATN-1, either through a bacterial conjugate (such as a PLP-binding bacterial protein) or it is equally possible that *C. elegans* does not directly require vitamin B6, but instead it relies on specific rate-limiting molecules from *E. coli* that require vitamin B6 for their synthesis (Figure 6G).”. We also revised the model based on the updated results (Figure 6G).

Minor

Line 1 is required?

Thank you for this suggestion. We added “is” in line 1 which is the title.

L26 and L222 should be protein - DAF-16.

Thank you for pointing this out. We corrected “daf-16” to “DAF-16” in line 26 and line 239.

L144 nutrient signals, nutrients, signals, or all of them. I actually think the distinction is important and in this case all could be involved?

Thank you for pointing this out. The distinction between nutrient signals, nutrients, signals is important. In our manuscript, we meant to discover the genetic components that is important to worm growth when fed *L. plantarum*. In the following section, we seek the nutrient signals and signaling pathways that could be involved. In line 158, they are “nutrients” in the context, thus we delete “signals”.

L222 *aak-2* should be in italic.

Thank you for pointing this out. We corrected “aak-2” to “*aak-2*” in line 222.

L227 *tatn-1* should be in italic.

Thank you for pointing this out. We corrected “tatn-1” to “*tatn-1*” in line 369.

L262 should the conclusion be part of the figure legend – I would delete it and only mention it in the main text.

Thank you for pointing it out. We deleted this conclusion sentence “indicating that heat-stable, UV stable, and non-secreted factors from *E. coli* are required for normal worm growth” in line 262.

L441 *act-1* should be in italic.

Thank you for pointing this out. We corrected “act-1” to “*act-1*” in line 488.

Figure 5A : the horizontal pdxH line is missing arrow heads.

Thank you for pointing this out. We added the arrow head in Figure 5A.

L440 References: *C. elegans*, worm genes etc. should consistently be in italic.

Thank you for pointing this out. We have carefully checked and corrected the worm and gene names in the references.

L 436 Acknowledging the CGC will help maintain their funding. GCG asks to include the following statement: "Some strains were provided by the CGC, which is funded by NIH Office of Research Infrastructure Programs (P40 OD010440)."

Thank you for bringing this to our attention. We benefited a lot from CGC by getting all the strains from their center. We agree that acknowledging CGC is important. We have added “*C. elegans* strains were provided by the Caenorhabditis Genetics Center, which is funded by NIH Office of Research Infrastructure Programs (P40 OD010440).” to the acknowledgments in lines 512-513.

Anders Olsen

Reviewer #2 (Remarks to the Author):

This manuscript by Feng et al., shows the effect on *C. elegans* growth of a diet of *L. plantarum*. The author convincingly show that traces amount of *E. coli* can supplement the nutritional deficit of this diet, and point to vitamin B6 as a key metabolite for postembryonic development. The approach, using an *E. coli* KO library is very useful and provide solid data on the pathway involved in the developmental defect. While the manuscript shows interesting data, the interpretation and claims of the authors are, at times, beyond the experimental evidence. For publication of this manuscript, my recommendation is to complete a couple of key experiments, or otherwise adapt the claims in the text.

Major concerns

- Relative to the nature of the developmental effect

Line 76: "...whereas worms fed solely on *L. plantarum* and developmentally arrested at early larval stage". The authors imply arrest of development, but this was not proven. It is difficult to differentiate arrest from developmental delay to the naked eye. However, the authors could use fluorescent reporters to analyze the state of the first divisions of postembryonic development, like seam cells and M cell. This way, it would be possible to assess if the observed effect corresponds to developmental arrest at a specific stage.

Thank you for your suggestion. We conducted new experiments using M lineage patterning as an indicator of worm development progression using *hlh-8::gfp* reporter expressed in the M cell. The result indicated that the development of worms fed with *L. plantarum* arrested at late L1 to L2 stage. We have added the results in Figure 1E and added the following description in lines 91-102 "To further elucidate the stage at which developmental arrest occurred in the presence of *L. plantarum*, we assessed M lineage patterning as an indicator of worm development progression. In postembryonic development, a single mesodermal blast cell (M) undergoes division to produce a small number of additional mesodermal cells. In hermaphrodites, the M divisions occurring in early larval development result in the formation of 14 striated body wall muscles, two sex myoblasts (SMs), and two coelomocytes. By the L4 stage, the SMs undergo division, resulting in 16 SM descendants located near the vulval opening. Here, we used *hlh-8::gfp* reporter expressed in the M-cell as the indicator. We observed that the GFP signal displayed the pattern characteristic of the stage between the late L1 stage to L2 stage in worms fed *L. plantarum* after 3 days, while worms fed OP50 exhibited a GFP signal pattern resembling the L4 stage (Figure 1E). This observation strongly suggests that the development of worms fed with *L. plantarum* arrested or stalled at the late L1 stage to L2 stage."

Line 22: "... the downstream metabolite pyridoxal 5-P (PLP, Vitamin B6) as essential nutritional factors initiating *C. elegans* postembryonic development" Why talk about initiation of development? Again, since it is not clear what is the stage the of the larvae on the *L. plantarum* diet (or the *pdx* mutants), it is difficult to confirm that Vit B6 is needed for initiating the process of postembryonic development. In order to state that Vit B6 is necessary for initiation of

development it is crucial to show that larvae are arrested before the first divisions of postembryonic development.

Thank you for pointing this out. We talked about initiation of development because we thought the worms could be arrested at L1 stage when we wrote the draft. To avoid misleading the reader, we used M lineage patterning as an indicator of worm developmental stage. The new result suggested the worms arrested at late L1 stage to L2 stage. We have changed “initiation of development” to “development” throughout the manuscript.

Figure 3B shows that larvae on the *L. plantarum* diet are larger than those on small amount of *E. coli*. In the same direction, Line 103: “the worms stayed at L1/L2 stage 5 days after been placed on the *L. plantarum* lawn”. Here, the authors refer to L1/L2, is this different from the stage that is found the starved L1s are used to initiate the experiments? Figure S2, bottom left picture: There are some larvae from the following generation, which are likely newly hatched L1. In the rest of the pictures, larvae are quite larger than those, indicating some growth and probably other developmental events took place.

Thank you for pointing this out. Yes, the worms fed *L. plantarum* are slightly bigger than starved L1s. Per your suggestion, we used M cell division to assess the stage of those arrested worms. The result indicated the development of worms fed with *L. plantarum* arresting at the L2 stage. We have added the results in Figure 1E and added the following paragraph in lines 91-102 “To further elucidate the stage at which developmental arrest occurred in the presence of *L. plantarum*, we assessed M lineage patterning as an indicator of worm development progression. In postembryonic development, a single mesodermal blast cell (M) undergoes division to produce a small number of additional mesodermal cells. In hermaphrodites, the M divisions occurring in early larval development result in the formation of 14 striated body wall muscles, two sex myoblasts (SMs), and two coelomocytes. By the L4 stage, the SMs undergo division, resulting in 16 SM descendants located near the vulval opening. Here, we used *hlh-8::gfp* reporter expressed in the M-cell as the indicator. We observed that the GFP signal displayed the pattern characteristic of the stage between the late L1 stage to L2 stage in worms fed *L. plantarum* after 3 days, while worms fed OP50 exhibited a GFP signal pattern resembling the L4 stage (Figure 1E). This observation strongly suggests that the development of worms fed with *L. plantarum* arrested or stalled at the late L1 stage to L2 stage.”

- Relative to the effect of PLP.

PLP on its own does not restore growth, alive *E. coli* is also needed. This means the effect on development is not directly depending on PLP. Actually, the authors state this in line 185 “suggesting that the beneficial role of PLP for larval growth is likely dependent on some downstream bacterial usage of PLP.” However, in the model in Figure 6, and in further sections of the manuscript, the authors seem to imply a direct role of PLP. This should be clarified in the text. Thank you for your suggestion. We agree with the reviewer. From Figure 5E, we concluded PLP on its own does not restore growth - alive *E. coli pdxH* mutant is also needed. To make it clearer, we changed the sentence to “these results demonstrate that *E. coli pdxH* mutant provides limited nutrition, with vitamin B6 deriving from its prior culturing conditions, supporting worm growth.” in lines 211-213. In this section, we also included “We found that supplementing 1mM PLP without live *pdxH* mutant did not promote the worm growth (Figure 5E), suggesting that the

beneficial role of PLP for larval growth either requires some live *E. coli* molecules to deliver the PLP to *C. elegans*, or *C. elegans* is dependent on some downstream bacterial usage of PLP.” in lines 197-201, “The *pdxH* mutant was grown in rich nutrient containing yeast extract, which provided the vitamins required for the mutant’s enzyme functions.” in lines 203-204.

This becomes especially relevant for the last part of the paper. In line 225, the authors say: “Previous studies have shown that TAT is a PLP-dependent enzyme that initiates the catabolism of tyrosine”. From there, they hypothesized that the effect of PLP on development could be mediated by the activation of DAF-16 by low levels of TAT, via AAK-2 activation. However, given that the authors proved that supplementation with PLP is not sufficient for growth, this is not a strong hypothesis. While the *pdxH* mutant shows low levels of *tant-1*, the authors do not prove that a reduction in *tant-1* has the same effect on growth. Furthermore, they do not show any experiments to support a role of AAK-2 or DAF-16 in the effect of the *L. plantarum* diet.

Thank you for pointing this out. In order to demonstrate the role of *daf-16* in the effect of the *L. plantarum* diet and raise our hypothesis, we did more experiments by using worm strain TJ356 (*daf-16::gfp*) to investigate DAF-16 translocation when worms are fed *L. plantarum* only, *L. plantarum* with wt *E. coli* (BW25113), and *L. plantarum* with *pdxH* mutant. The results are showing in Figure 6E, indicating the increased DAF-16 nuclear translocation when fed *L. plantarum*, including “To shed light on the participation of DAF-16 in worm developmental arrest mediated by *L. plantarum*, we investigated the subcellular localization of DAF-16 in worms exposed to various conditions, including those exclusively fed *L. plantarum*, *L. plantarum* in combination with 0.2 μ l of BW25113, *L. plantarum* with 0.2 μ l of the *E. coli pdxH* mutant, and *L. plantarum* with 0.2 μ l of the *E. coli pdxH* mutant in addition to 1 mM PLP. We found that worms fed *L. plantarum* only or *L. plantarum* with 0.2 μ l of the *E. coli pdxH* mutant exhibited a significant increase in the nuclear localization of DAF-16 after 4 days incubation when compared to worms fed *L. plantarum* with 0.2 μ l of wild-type *E. coli* BW25113 (Figure 6E). In addition, PLP supplementation reversed the DAF-16 nuclear translocation (Figure 6E), indicating the nuclear translocation of DAF-16 in worms fed *L. plantarum* with 0.2 μ l of the *E. coli pdxH* mutant is likely due to stress associated with deficiency in PLP or a decrease in a downstream PLP-regulated process.” in line 241-251.

In addition, we investigated the growth of *daf-16 (mu86)* null worm mutant on the *L. plantarum* diet. Result is included in Figure 6F, “To further clarify the role of *daf-16*, we asked whether *daf-16(mu86)* null mutants developmentally arrest or can develop on *L. plantarum*. When this mutant was exclusively fed *L. plantarum*, the worms exhibited developmental arrest (Figure 6F). This suggests that processes that lead to developmental arrest do not rely on *daf-16* for activation.” In line 251-255. Therefore, we revised the discussion based on those new results, including “In this study, although DAF-16 does translocate into the nucleus upon *L. plantarum* feeding, it does not seem to be the primary cause of developmental arrest or stalling. The developmental defect from the *L. plantarum* diet is possibly due to a lack of rate-limiting factor crucial for robust developmental progression. DAF-16 might just promote a coping mechanism to sustain worm survival until an adequate amount of rate-limiting substance is acquired. This rate limiting substance could potentially be vitamin B6 delivered via a bacteria-derived vehicle or other *E. coli* factors that rely on vitamin B6 for their synthesis.” In lines 410-417.

As for the hypothesis on *tatn-1*, we added further references showing knockdown of *tatn-1* gene expression can stall larval development {24,25} and moved this part to the discussion, including “Interestingly, our RNA-seq analysis revealed multiple PLP binding genes are decreased in larval *C. elegans* fed with *L. plantarum* (Figure 6C), one of which is *tatn-1* (Figure 6C). *tatn-1* gene encodes a tyrosine aminotransferase (TAT), previous studies have shown that TAT is a PLP-dependent enzyme catalyzes conversion of tyrosine to 4-hydroxyphenylpyruvate, which is a rate-limiting step to metabolize tyrosine to fumarate and acetoacetate^{22,23}. RNAi knockdown of *tatn-1* gene expression can stall larval development^{24,25}. Here, we hypothesize that the PLP from *E. coli* functions as a coenzyme of worm TAT, the absence of PLP from the dietary bacteria leads to a reduced expression of *tatn-1* (Figure 6C), and subsequently leading to downstream deficiency in fumarate and acetoacetate or detrimental buildup of tyrosine and the developmental arrest. It is possible that *E. coli* is responsible for delivering vitamin B6 to *C. elegans* TATN-1, either through a bacterial conjugate (such as a PLP-binding bacterial protein) or it is equally possible that *C. elegans* does not directly require vitamin B6, but instead it relies on specific rate-limiting molecules from *E. coli* that require vitamin B6 for their synthesis (Figure 6G).” in lines 368-380.

Line 189: “The results show that the lack of *pdxH* does not affect worm growth (Figure 5F).

Do *pdxH* mutants completely lack Vit B6? Could *pdxH* mutants be making PLP from PL?

Do other *pdx* mutants also lead to growth when used in high amounts?

Thank you for bringing this to our attention. Lack of *pdxH* does not affect worm growth with *E. coli* as the diet alone, which means that the function of *pdxH* can be compensated for by increasing the amount of the *E. coli* mutant. It could be some salvage pathway that makes PLP from PL. To stress the role of PLP from *E. coli* that promotes *L. plantarum* usage, we included “The *pdxH* mutant was grown in rich nutrient containing yeast extract, which provided the vitamins required for the mutant’s enzyme functions. Taken together, these results demonstrate that *E. coli pdxH* mutant provides limited nutrition, with vitamin B6 deriving from its prior culturing conditions, supporting worm growth.” in lines 210-213. “Lack of autonomous *E. coli* synthesis of PLP does not affect worm growth as the diet alone, provided sufficient live mutant bacteria is fed to the worms (Figure 5F). This observation suggests that PLP-binding enzymes in living *pdxH* mutants can either (i) serve as a delivery vehicle for the vitamin to *C. elegans* enzymes, such as tyrosine amino transferase, or (ii) vitamin B6 might be required for *E. coli* PLP-dependent enzymes to synthesize essential molecules, such as amino acids, that might be deficient in the context of a *L. plantarum* diet for larval *C. elegans*³⁰. When *pdxH* mutant is cultivated in nutritionally rich LB broth (usually containing yeast extract), which likely provides vitamin B6 along with all of the amino acids that are synthesized by vitamin B6 binding enzymes. It’s possible that the essential enzymes in *pdxH* mutant diet are loaded with vitamin B6 from its yeast extract containing growth media. Consequently, the greater the quantity of *pdxH* mutant added to *C. elegans*, the more indirectly yeast-derived vitamin B6 is introduced, thereby supporting worm growth.” in lines 388-400 in the discussion.

The RNA-seq results showed fluctuated gene transcription in host PLP binding.

- Relative to the role of DAF-16.

Line 25: “Additionally, bacterial PLP may act as a cofactor for host tyrosine aminotransferase, thereby promoting the translocation of daf-16 to nucleus.” This sentence in the abstract suggests that the authors have analysed DAF-16 localization, which is not the case.

Thank you for pointing this out. We revised the abstract based on new results, includes “Additionally, the developmental arrest induced by the *L. plantarum* diet in worm does not depend on the activation of FoxO/DAF-16.” in lines 25-26.

Lines 219-231: This paragraph belongs to the Discussion, not the results section.

Thank you for this suggestion. We moved this paragraph to the discussion.

Furthermore, the interpretation by the author of the cited manuscript on tant-1 seems to be wrong. The cited manuscript concludes, from the results in Figure 6E: “Together these findings suggest that the tant-1 enhancement of the eak dauer formation phenotype could be due to an increase in daf-16 transcriptional activity without an accompanying significant change in DAF-16 subcellular localization. The model and figure legend 6E state: “A proposed model suggests that bacterial PLP acts as a cofactor of host TATN-1, promoting worm growth through DAF-16 translocation”. Does this mean translocation to the nucleus? This contradicts what is stated at the end of the results section and does not make sense in terms of what we know about DAF-16 activity. According to the model, it seems that PLP leads to reduction of tant-1, when it should be the opposite. This is way the model seems to propose that PLP leads to nuclear localization of DAF-16, when it should be its absence what provokes that effect. PLP is necessary to initiate development but then this sentence says that promotes translocation to the nucleus. Translocation of DAF-16 to the nucleus would arrest development, not promote it.

Thank you for pointing this out. In order to demonstrate the role of *daf-16* in the effect of the *L. plantarum* diet and raise our hypothesis, we did more experiments by using worm strain TJ356 (*daf-16::gfp*) to investigate DAF-16 translocation when worms are fed *L. plantarum* only, *L. plantarum* with wt *E. coli* (BW25113), and *L. plantarum* with *pdxH* mutant. The results are showing in Figure 6E, indicating the increased DAF-16 nuclear translocation when fed *L. plantarum*, including “To shed light on the participation of DAF-16 in worm developmental arrest mediated by *L. plantarum*, we investigated the subcellular localization of DAF-16 in worms exposed to various conditions, including those exclusively fed *L. plantarum*, *L. plantarum* in combination with 0.2 µl of BW25113, *L. plantarum* with 0.2 µl of the *E. coli pdxH* mutant, and *L. plantarum* with 0.2 µl of the *E. coli pdxH* mutant in addition to 1 mM PLP. We found that worms fed *L. plantarum* only or *L. plantarum* with 0.2 µl of the *E. coli pdxH* mutant exhibited a significant increase in the nuclear localization of DAF-16 after 4 days incubation when compared to worms fed *L. plantarum* with 0.2 µl of wild-type *E. coli* BW25113 (Figure 6E). In addition, PLP supplementation reversed the DAF-16 nuclear translocation (Figure 6E), indicating the nuclear translocation of DAF-16 in worms fed *L. plantarum* with 0.2 µl of the *E. coli pdxH* mutant is likely due to stress associated with deficiency in PLP or a decrease in a downstream PLP-regulated process.” in line 241-251.

In addition, we investigated the growth of *daf-16 (mu86)* null worm mutant on the *L. plantarum* diet. Result is included in Figure 6F, “To further clarify the role of *daf-16*, we asked whether *daf-16(mu86)* null mutants developmentally arrest or can develop on *L. plantarum*. When this mutant

was exclusively fed *L. plantarum*, the worms exhibited developmental arrest (Figure 6F). This suggests that processes that lead to developmental arrest do not rely on daf-16 for activation.” In line 251-255. Therefore, we revised the discussion based on those new results, including “In this study, although DAF-16 does translocate into the nucleus upon *L. plantarum* feeding, it does not seem to be the primary cause of developmental arrest or stalling. The developmental defect from the *L. plantarum* diet is possibly due to a lack of rate-limiting factor crucial for robust developmental progression. DAF-16 might just promote a coping mechanism to sustain worm survival until an adequate amount of rate-limiting substance is acquired. This rate limiting substance could potentially be vitamin B6 delivered via a bacteria-derived vehicle or other *E. coli* factors that rely on vitamin B6 for their synthesis.” In lines 410-417.

As for the hypothesis on *tatn-1*, we added further references showing knockdown of *tatn-1* gene expression can stall larval development {24,25} and moved this part to the discussion, including “Interestingly, our RNA-seq analysis revealed multiple PLP binding genes are decreased in larval *C. elegans* fed with *L. plantarum* (Figure 6C), one of which is *tatn-1* (Figure 6C). *tatn-1* gene encodes a tyrosine aminotransferase (TAT), previous studies have shown that TAT is a PLP-dependent enzyme catalyzes conversion of tyrosine to 4-hydroxyphenylpyruvate, which is a rate-limiting step to metabolize tyrosine to fumarate and acetoacetate^{22,23}. RNAi knockdown of *tatn-1* gene expression can stall larval development^{24,25}. Here, we hypothesize that the PLP from *E. coli* functions as a coenzyme of worm TAT, the absence of PLP from the dietary bacteria leads to a reduced expression of *tatn-1* (Figure 6C), and subsequently leading to downstream deficiency in fumarate and acetoacetate or detrimental buildup of tyrosine and the developmental arrest. It is possible that *E. coli* is responsible for delivering vitamin B6 to *C. elegans* TATN-1, either through a bacterial conjugate (such as a PLP-binding bacterial protein) or it is equally possible that *C. elegans* does not directly require vitamin B6, but instead it relies on specific rate-limiting molecules from *E. coli* that require vitamin B6 for their synthesis (Figure 6G).” in lines 368-380.

Minor and typos:

- Relative to the effect of *L. plantarum*

From <https://doi.org/10.3389/fnut.2022.1031502>

“Some species of microbiota lack the ability to biosynthesize vitamin B6, such as most genera within the Firmicutes phylum (*Veillonella*, *Ruminococcus*, *Faecalibacterium*, and *Lactobacillus* spp.)”.

Thank you for this suggestion. We add this to the discussion “Some species of microbiota including *Lactobacillus* lack the ability to biosynthesize vitamin B6, which makes synergistic in providing nutrients to the host important between microbes.” in lines 383-385.

Line 131: “Thus, *E. coli* factors that rendered the *L. plantarum* edible are not secreted, heat nor UV stable.” However, in the model in Figure 6, PLP seems to be secreted from the bacteria.

Thank you for pointing this out. As indicated in the rightmost image in Figure 3C, worms can not grow with *E. coli* conditioned media, thus we concluded the factor that rendered the *L. plantarum*

edible are not secreted. To avoid confusion, we modified the arrow origin in Figure 6G from outside to inside the bacteria.

- Table 1 is not available for review.

Thank you for pointing this out. We uploaded the Table 1.

- The number of independent replicates of the experiments is not stated in the text or legends.

Thank you for pointing this out. We added the number of replicate experiments and n in the individual experiments in the figure legends. To be specific, "All data are representative of at least three independent experiments. n = number of worms scored. Data are represented as mean \pm SEM. * indicates P-value < 0.05, ** indicates P-value < 0.01, N.S. indicates non-significant difference." are included in lines 268-270, lines 280-281, lines 295-297, lines 320-322, and lines 334-336.

- Others:

Line 26: daf-16 should be DAF-16

Thank you for pointing this out. We changed "daf-16" to "DAF-16" in line 26.

Line 73: Revise the grammar of the sentence: "To investigated the worm growth on Gram-positive bacterium *L. plantarum*, a commonly consumed probiotic strain, and compare that with standard laboratory food *E. coli* OP50."

Thank you for pointing this out. We revised this sentence to "To investigate the effect of two different food sources, a commonly consumed probiotic Gram-positive bacterium *L. Plantarum* and the standard laboratory food *E. coli* OP50, we fed synchronized L1 worms with *L. plantarum* and *E. coli* OP50 individually and measure worm development." in line 73-75.

Line 76: Remove the "and at the end of the line.

Thank you for the suggestion. We removed the "and" in line 76.

Line 222: aak-2 and daf-16 should be AAK-2 and DAF-16

Thank you for pointing this out. We changed "aak-2" and "daf-16" to "AAK-2" and "DAF-16" in line 239.

Line 222: "has a positive effect on DAF-16". Please state what is the effect.

Thank you for pointing this out. We revised this to "activates DAF-16".

Line 227: tant-1 should be italicized

Thank you for pointing this out. "*tatn-1*" is now italic in line 369.

Line 365: "Then" should be "The"

Thank you for pointing this out. We changed "Then" to "The" in line 365.

Figure 3A. There is a typo on the Y axis legend mm3

Thank you for pointing this out. We changed "mm³" to " μm^3 " in Figure 3A

Figure 4B. What is “WO”

Thank you for pointing this out. This should be a no bacteria feeding control. We changed it to “NO” in Figure 4B.

Figure 5A. Arrowhead missing in the line between PLP and PMP.

Thank you for pointing this out. The *pdxH* is necessary for PMP conversion to PLP. We added the arrow head in Figure 5A.